# Assessment of Iran's Mangrove Forest Dynamics (1990–2020) Using Landsat Time Series

Yousef Erfanifard [1] , Mohsen Lotfi Nasirabad [1] and Krzysztof Stereńczak [2,*]

1    Department of Remote Sensing and GIS, Faculty of Geography, University of Tehran, Tehran 1417853933, Iran
2    Department of Geomatics, Forest Research Institute, 05-090 Sękocin Stary, Poland
*    Correspondence: k.sterenczak@ibles.waw.pl; Tel.: +48-22-7150-325

**Abstract:** Mangrove forests distributed along the coast of southern Iran are an important resource and a vital habitat for species communities and the local people. In this study, accurate mapping and spatiotemporal change detection were conducted on Iran's mangroves for three decades, using the Landsat imagery available for the years 1990, 2000, 2010, and 2020. Four general vegetation indices and eight mangrove-specific indices were employed for mangrove mapping in three study sites. Additionally, six important landscape metrics were implemented to quantify the spatiotemporal alteration of the mangrove forests during the study period. Our results showed the robustness of the submerged mangrove recognition index (SMRI), validated as the most effective index (F1-score $\geq$ 0.89), which was used for mangrove identification within all nine sites. The mangrove area of southern Iran was estimated at approximately 13,000 ha in 2020, with an overall increase of 2313 ha over the whole period. A similar trend could be observed for both the landscape connectivity and complexity. Our results revealed that a stronger connectivity and higher complexity could be detected in most sites, while there was increased fragmentation and a weaker connection in some locations. This study provides an accurate map of Iran's mangrove forests over time and space.

**Keywords:** mangrove-specific index; SMRI; landscape metric; NDVI; change detection

## 1. Introduction

Mangrove forests are one of the important habitats in the coastal regions of tropical and subtropical parts of the world [1,2]. They are one of the most productive ecosystems that provide a unique habitat for many terrestrial and marine species and valuable goods and services for local communities. Additionally, mangroves not only play a key role in the conservation of coastal areas and intertidal regions from environmental hazards but also participate in ecosystem services such as carbon sequestration [3–5]. Murdiyarso et al. [6] showed that the annual rate of carbon storage in mangroves is two to four times greater than that of tropical forests. Despite their ecological and socioeconomic importance, the global loss rate of mangrove forests has reached 0.2–0.7% per annum. As reported by FRA 2020, the area of mangroves globally decreased more than 10,000 km$^2$ between 1990 and 2020 [1,7–9]. Climate change and human activities are known as major threats for mangrove forests, although investigations have shown that land-use change, through conversion to aquaculture, rice, and palm plantations, is the most important driver of deforestation [10–12]. Effective management strategies and socioeconomic policies are required to reverse the trend of global mangrove loss and ensure their conservation for future generations [13,14].

Mangrove ecosystems consist of shrubs and trees that mostly grow in the intertidal zones along most tropical shorelines between latitudes 30°N and 30°S. Nevertheless, the global spatial distribution of mangroves was probably more extensive during the warm climate of the Eocene epoch, as pollens of genus *Avicennia* were found above latitude 72°N (present-day Siberia). The establishment and growth of these elements of warm tropical

terrestrial flora were limited to lower latitudes during the Oligocene–Miocene transition, when temperatures significantly decreased in high altitudes. Mangroves were limited to tropical and subtropical latitudes during the Holocene until the present, mainly because of climate and sea-level fluctuations. In contrast to our knowledge about the spatial extent of mangroves across geological epochs, the total area changes of the forests remained unknown until the 1980s, when remote sensing science and technology facilitated their accurate mapping at local and global scales. Mangrove habitats are established in intertidal zones that may frequently be submerged during local high tides. Moreover, mangrove trees thrive on muddy islands generally surrounded by waterways, and their roots are submerged in water. Therefore, traditional methods of surveying are not efficient to map the spatial distribution of mangroves, while remotely sensed data can provide reliable mapping approaches as an alternative to expensive and time-consuming field-based methods. The first investigation on total area of global mangrove forests based on remote sensing, by Giri et al. [15], showed that the global mangrove extent was 137,760 km$^2$ in 2000. However, the findings of FRA 2020 showed global mangrove areas were approximately 152,900 and 147,170 km$^2$ in 2000 and 2020, respectively [1,9,14,16].

Mangrove forests form pure and mixed stands along the shorelines of the Persian Gulf and Gulf of Oman, which cover almost 19,700 ha. Among the six countries of the region (i.e., Iran, United Arab Emirates, Oman, Saudi Arabia, Qatar, Bahrain), Iran ranks first in total mangrove area (approximately 47%) [14,17,18]. Mangrove spatial distribution is one of the first quantitative characteristics influenced by natural and anthropogenic threats. Therefore, extent monitoring turns to be a reliable approach to acknowledge the historical changes and current status of mangroves. Moreover, efficient conservation planning and sustainable management require accurate maps of mangroves that show their spatiotemporal changes.

The current literature on mangrove forests along the coast of southern Iran lacks a long-term analysis of the dynamics and an accurate area estimation using robust remote sensing techniques, although the spatial distribution and temporal changes of mangroves have been reported in a limited number of studies [19,20]. For instance, Makowski and Finkl [14] reported that the spatial extent of mangrove forests in Persian Gulf and Gulf of Oman was mapped from 1977 to 2017 using the Normalized Difference Vegetation Index (NDVI) on Landsat imagery. They found that Iran's mangrove forests are spatially located in 11 sites and that their total area increased from 1977 (4735 ha) to 2017 (9403 ha). In another investigation, FRA 2020 mentioned that the total area of Iran's mangrove forests was 19,230 ha in 2020. The FRA 2020 country report indicated that the mangroves decreased from 25,760 to 19,230 ha between 1990 and 2020 [9]. There are discrepancies in the area estimation of mangrove forests, mainly due to different remote sensing data, vegetation indices, and classification algorithms. Gandhi and Jones [21] found that the most common classification error in their study was the assignment of mangrove pixels to the water class, particularly within <900 m$^2$ patches. Consequently, it seems necessary to update Iran's mangrove's present status and long-term dynamics, based on more efficient processing approaches.

Mangrove forest canopies exhibit unique spectral signatures on remote sensing imagery. Some authors have demonstrated the difference among the spectral reflectance patterns of mangroves and other land cover types including terrestrial vegetation, especially within the ranges corresponding to near infrared (NIR) and short-wave infrared (SWIR) wavelengths [2,7,22]. Giri [23] mentioned that a combination of red (0.63–0.69 μm), NIR (0.77–0.90 μm), and two SWIR (1.55–1.75 μm, 2.09–2.35 μm) wavelengths derived from Landsat data are appropriate to detect mangroves. Additionally, Baloloy et al. [22] demonstrated that the difference in the spectral responses between mangroves and terrestrial vegetation in red band and NIR are significantly smaller than the difference observed in SWIR. Despite their distinct spectral characteristics, individual bands of remotely sensed data are not appropriate for mangrove forest survey. Therefore, various forms of vegetation indices (VIs), defined as the spectral transformation of two or more bands, are the widely

used approach to map mangrove forests. Previous studies on mangrove forests have mostly used NDVI, which is calculated from red band and NIR [12,19,24]. However, a number of researchers have illustrated that NDVI and other similar VIs, such as the Soil-Adjusted Vegetation Index (SAVI) and leaf area index (LAI), are not specifically designed to discriminate mangroves from other land cover types, e.g., terrestrial vegetation. It was also found that the VIs such as NDVI and SAVI are not efficient in the distinction of mangroves, since submerged mangrove forests show very low values of the indices, which can result in their misclassification [25,26]. It seems difficult to obtain clear results on the performance of commonly used VIs in mangrove mapping and monitoring.

To address the challenge of mangrove discrimination on remote sensing imagery, some mangrove-specific indices (MSIs) have been developed by various scientists. One of the first MSIs, proposed by Winarso et al. [27], was the Mangrove Index (MI) using NIR and SWIR bands (i.e., bands 5 and 6) derived from Landsat 8 data. Baloloy et al. [22] recently suggested Mangrove Vegetation Index (MVI) with the similar objective of mangrove mapping. MSIs are not limited to MI and MVI, and structurally different indices (e.g., Normalized Difference Mangrove Index (NDMI), Combined Mangrove Recognition Index (CMRI) have been developed in the literature [7,25,28]. To the best of our knowledge, MSIs have rarely been compared with each other or widely used vegetation indices to explore their robustness. Furthermore, their efficiency has mostly been reported within the habitats that they were designed for. In a comprehensive investigation, Xia et al. [26] evaluated the performance of two MSIs (i.e., Mangrove Recognition Index (MRI), Submerged Mangrove Recognition Index (SMRI)) and four VIs (i.e., NDVI, SAVI, Ratio Vegetation Index (RVI), Enhanced Vegetation Index (EVI)). Results showed the performance of the MSIs versus the four VIs, although SMRI could detect mangroves with higher accuracy compared to MRI. In general, a few comparative studies have investigated the efficiency of VIs and recently developed MSIs; however, the current literature lacks a comprehensive and critical review of general VIs and MSIs.

Considering the findings of previous studies in the international and local literature, we hypothesized that the total area of Iran's mangrove forests has increased over the last three decades. To test the hypothesis, the spatial extent of mangroves within each of the sites is mapped between 1990 and 2020 on Landsat imagery, to quantify the dynamic changes of the forests through a comparison of the otal areas and an analysis of the landscape metrics. We also hypothesized that the performance of the diverse existing MSIs designed for Landsat data in a new mangrove forest is different from their accuracies in the study sites that they were developed for. We tested the hypothesis to find out the robustness of MSIs and VIs and reveal their applicability and limitations at international scales, within habitats different from where they were constructed and tested. The performance of recently developed MSIs is also investigated to select the most accurate one for Iran's mangrove mapping. The results may provide information on national estimates and the dynamics of mangrove forests in southern Iran, based on a reliable procedure. Additionally, this study is a step toward the practical application of the indices specifically designed for mangrove forest survey on remote sensing imagery and a demonstration of their true strength versus other VIs in similar circumstances.

## 2. Materials and Methods

### 2.1. Study Area

Mangrove forests along the southern coast of Iran are located between the latitudes 25°10′ to 27°55′ north and longitudes 51°25′ to 61°25′ east. The forests are a unique natural beauty in Iran and potential sites for ecotourism and recreation. Significantly dense mangroves are observed in Qeshm Island and Khamir Harbor, while they form dense and open stands within other sites. Iran's mangroves comprise two species, i.e., grey mangrove (*Avicennia marina* (Forssk.) Vierh.) and loop-root mangrove (*Rhizophora mucronata* Lam.). Grey mangrove is the dominant species forming pure stands, while loop-root mangrove can be observed only in mixed stands of both species in Sirik and Jask (S6 and S7 in Hormozgan

Province, Figure 1). The mangroves are mainly threatened by natural (pest damage, storm, acid rain) and anthropogenic (oil spills, industrial wastes, infrastructure development) factors. However, establishment of protected areas in almost all mangrove sites is an effective management tool extensively used by the government to prevent their loss and degradation. In addition to restoration, afforestation and reforestation are two management strategies successfully undertaken since 1990s supported by local communities.

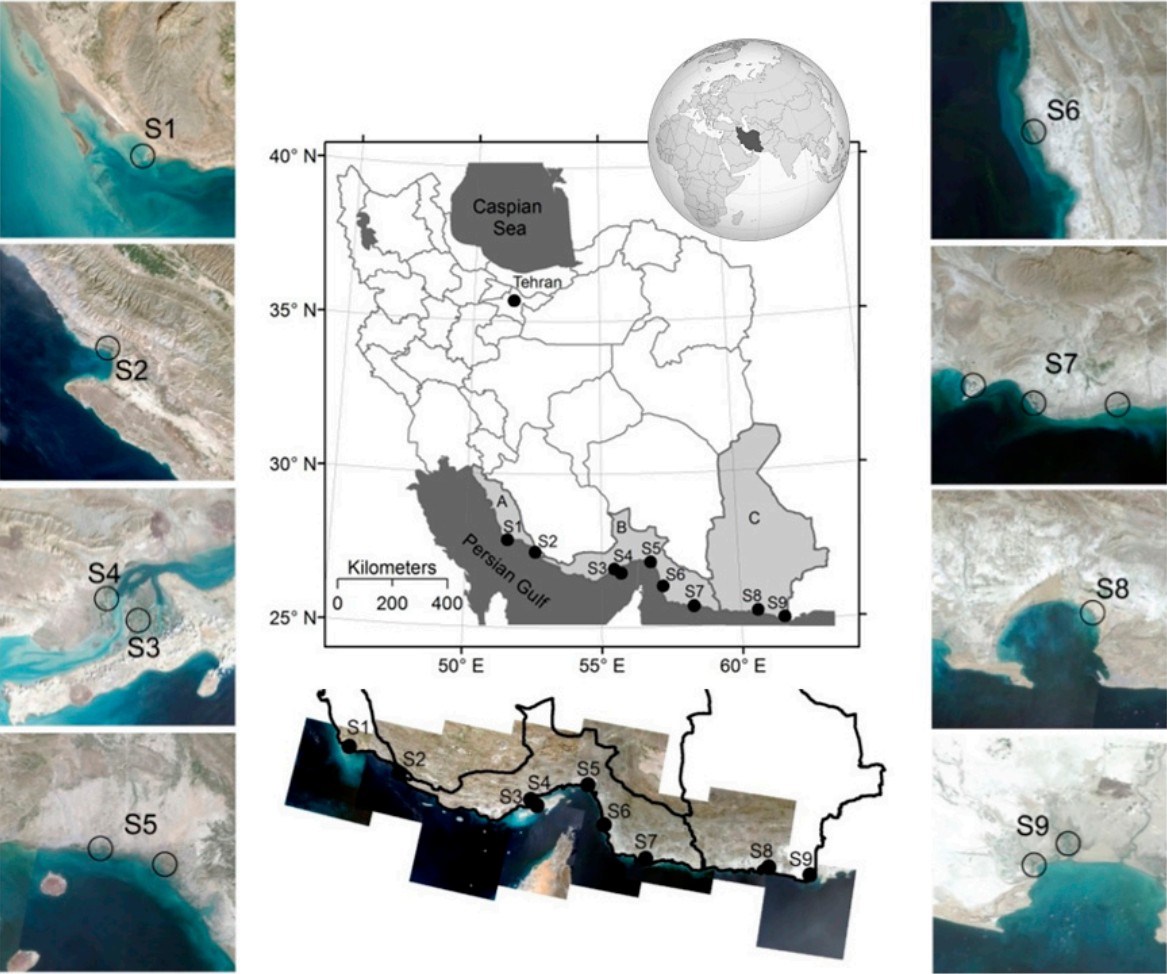

**Figure 1.** Spatial distribution of all major mangrove sites in Iran investigated in this study, i.e., S1 and S2 in Bushehr Province (**A**), S3–S7 in Hormozgan Province (**B**), and S8 and S9 in Sistan and Baluchestan Province (**C**). The sites were shown on Landsat 8 mosaic images (R = Band 3, G = Band 2, B = Band 1) collected in March 2020, with the overlay of administrative boundaries of Bushehr Province, Hormozgan Province, and Sistan and Baluchestan Province.

All mangrove sites in Persian Gulf and Gulf of Oman coasts were considered as the study sites, i.e., Mond protected area (S1) and Nayband National Park (S2) in Bushehr Province; Hara protected area (S3), Haraye Khamir protected area (S4), Hara Tiab and Minab protected area (S5), Hara Roud-e Gaz protected area (S6), and Hara-e Gabrik and Jask protected area (S7) in Hormozgan Province; and Chabahar hara forests (S8) and Bahookalat protected area (S9) in Sistan and Baluchestan Province (Figure 1). Some subsites are defined within each major mangrove site; however, they are considered as one site in this study. For instance, there are two different subsites within S5, i.e., the forests located at the mouth of the Shoor River and the forests distributed in Tiab Harbor estuary. Additionally, S7 has three subsites: Jask Harbour and Sourgalm and Gabrik villages. Hara Biosphere Reserve is one of 13 UNESCO biosphere reserves of Iran, registered in 1976, which consists of Qeshm Island (S3) and Khamir Harbor (S4).

The last area of assessment of mangrove forests in 2011 using NDVI on Landsat imagery showed that S1 (17.9 ha) and S3 (5542.7 ha) comprised the smallest and largest mangrove sites, respectively. Moreover, the total area of mangroves was 11,015.1 ha. The mangrove sites in Hormozgan Province supported 93.5% (10,305.2 ha) of the total mangroves in Iran. Meanwhile, the extent of mangroves in provinces of Bushehr and Sistan and Baluchestan was 1.4% (149.8 ha) and 5.1% (560.1 ha) of the total mangroves, respectively. According to results of the study, no mangrove stands were observed in S8 in 2011.

### 2.2. Satellite Data and Input Bands

The Landsat time series data covering all areas of interest (i.e., S1 to S9) were considered in the present study, as the Landsat archive is the longest continuous satellite-based record of Earth's land cover in existence, which was available for the study sites. The data were downloaded from the USGS (United States Geological Survey) EarthExplorer from 1990 to 2020, with 10-year lags (see Table A1 for the dates of the images). The cloud-free images of Landsat 5 TM (L5), Landsat 7 ETM+ (L7), and Landsat 8 OLI (L8) were controlled for topographic distortion by employing the digital terrain model (DTM) of each region, and, if required, the images were geometrically corrected using ground control points. Atmospheric correction was also performed using FLAASH (Fast Line-of-sight Atmospheric Analysis of Spectral Hypercubes) algorithm in ENVI.

For each mangrove site, Landsat data were obtained for high- and low-tide levels, based on visual interpretation, and the tidal data obtained from WXTide32 software [29] for the study years (i.e., 1990, 2000, 2010, 2020). We used the reference stations in Bushehr Province (Jazirat Kharg), Hormozgan Province (Jazirat Farur, Hengam Island, Bandar Abbas, Jask Bay), and Sistan and Baluchestan Province (Gwatar Bay). A single Landsat tile covered the subsites of a mangrove sites, e.g., S5, S7, and S9; however, S3 was covered by two Landsat tiles (Figure 1).

The Landsat (i.e., L5, L7, L8) images of the study sites required a series of preprocessing prior to being used to derive the vegetation indices. In the first step, radiometric corrections were conducted with application of dark object subtraction (DOS) method on the Landsat images, to reduce atmospheric effects. In the second step, the Landsat images were geometrically corrected using rational polynomial coefficients (RPCs), with at least 10 evenly dispersed ground control points within each image tile extracted from very high resolution (VHR) images of Google Earth, UltraCam airborne images, and 1:25,000 topographic maps. The average root-mean-square errors (RMSEs) were less than 0.5 pixels for all Landsat images in both the X and Y directions.

Based on the literature on spectral signature of mangrove forests and the indices used to detect them on Landsat imagery, the required spectral wavelengths were obtained to be used in computation of the VIs and MSIs investigated in the present study (Table 1). These spectral wavelengths were Blue (Band 1 in L5 and L7; Band 2 in L8), Green (Band 2 in L5 and L7; Band 3 in L8), Red (Band 3 in L5 and L7; Band 4 in L8), NIR (Band 4 in L5 and L7; Band 5 in L8), SWIR1 (Band 5 in L5 and L7; Band 6 in L8), and SWIR2 (Band 7 in L5, L7, and L8) (see Table A2 for more details). Bands of SWIR1 and SWIR2 are affected by leaf water content; however, studies have shown that SWIR1 region is more efficient in mangrove recognition from open and dense terrestrial vegetation compared to SWIR2 [7,22].

### 2.3. Computation of Vegetation Indices

The present study had two phases (Figure 2). The first phase involved a comparative assessment of widely used VIs and MSIs that have been developed to discriminate mangroves on L8 data for 2020. The assessment of the indices was performed within three mangrove sites, i.e., S2 (Bushehr Province), S6 (Hormozgan Province), and S9 (Sistan and Baluchestan Province) to identify the most efficient index in the study areas. In the second phase, the Landsat bands (Table A2) were used as inputs for computing the selected index for all mangrove sites (Figure 1) in all study years (i.e., 1990, 2000, 2010, 2020).

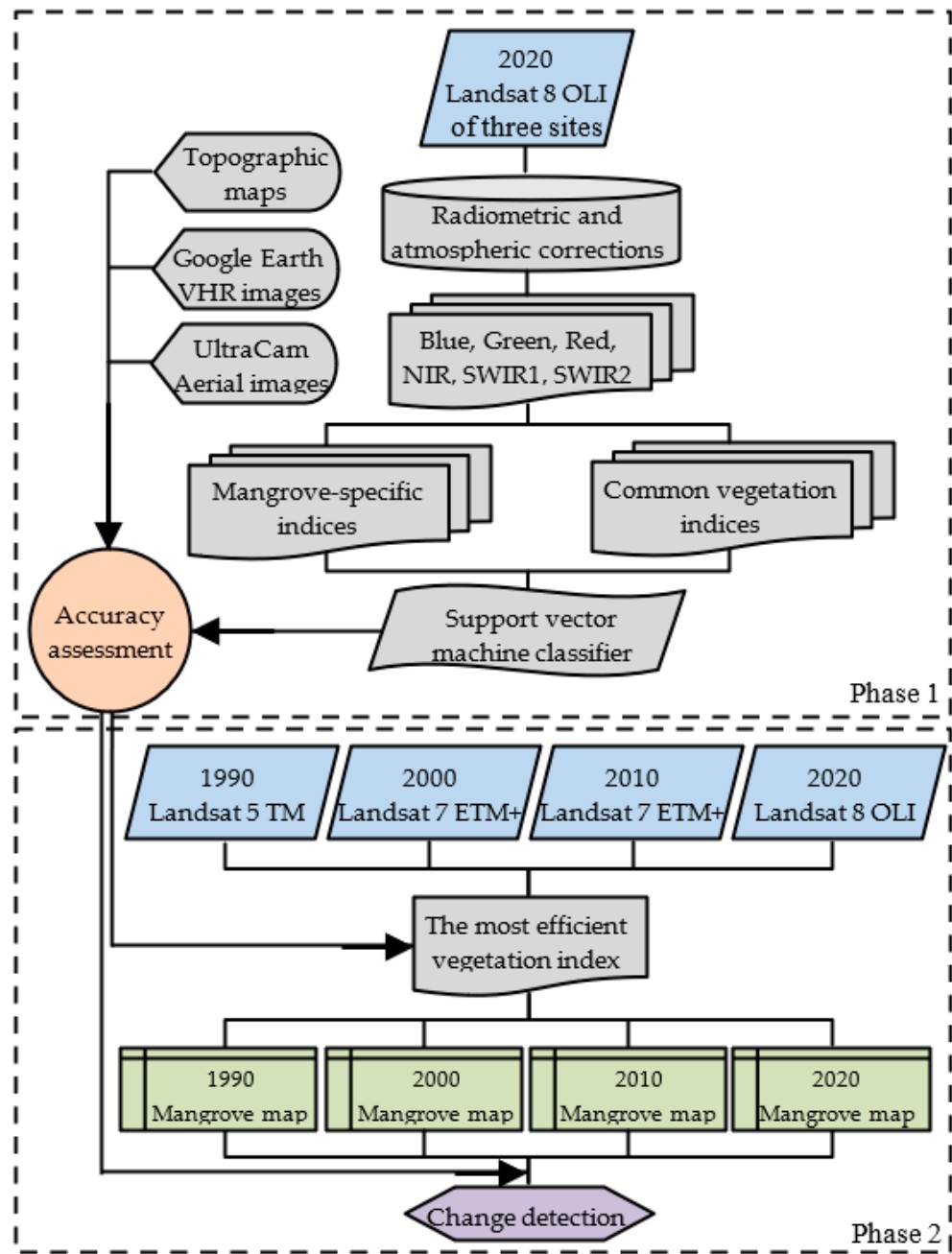

**Figure 2.** The methodological framework of Iran's mangrove forest assessment.

Previous studies have used a wide range of VIs for mangrove mapping on Landsat images, so four of them were considered in this study (Table 1). One of the most frequently used VIs is NDVI, introduced by Rouse et al. [30]. The index was successfully implemented to delineate green vegetation, including mangrove forests, from other land use/land cover classes on different remote sensing data [31]. For instance, Lovelock et al. [32] indicated that NDVI is significantly correlated with canopy loss in mangrove forests. NDVI can be effectively used to monitor degradation and deforestation of mangroves. In addition to NDVI, SAVI, proposed by Huete [33], is a popular index employed to classify mangrove and non-mangrove areas [7,26,34]. SAVI considers a soil-brightness correction factor (i.e., L, Table 1) to reduce the influence of soil brightness in open vegetation patches. As suggested by previous studies [32,35], the L factor is 0.75 in the present research. Normalized Difference Water Index (NDWI), developed by Gao [36], is strongly sensitive to vegetation water content. The index has successfully been used to delineate

mangrove stands on different remote sensing imagery [25,37]. Pastor-Guzman et al. [38] investigated the capability of NDWI in examining biophysical variables and tracking seasonality of mangroves. According to their achievements, it was revealed that NDWI time series is reliable for monitoring mangrove water stress, since NDWI has the ability to change in proportion to canopy moisture content. While NDVI has significant correlations with green biomass and canopy chlorophyll content, EVI is more sensitive to canopy structural characteristics such as canopy architecture and leaf area index [39,40]. In studying mangrove forests, EVI has been found to be a reliable index in classifying mangroves and non-mangrove areas [7,26].

There are diverse and multiple VIs (e.g., NDVI, SAVI, NDWI, EVI) that have been developed to discriminate vegetation and measure their temporal and spatial variations. However, these indices cannot specifically characterize mangroves from other vegetation types such as terrestrial forests. The investigation by Baloloy et al. [22] shows that the SWIR1 and SWIR2 bands are the region within which mangrove and terrestrial forests are recognizable, because of the difference in their spectral reflectance properties. Although SWIR1 and SWIR2 are helpful for mangrove and non-mangrove vegetation differentiation, the bands are not considered in formulation of common VIs. Meanwhile, some VIs may provide similar values in vegetation types such as mangrove forests and dense terrestrial vegetation. Therefore, indices have been developed utilizing Green, NIR, SWIR1, and SWIR2 wavelengths to specifically delineate mangroves considering their spectral behavior (Table 1).

The Mangrove Index (MI) was one of the first mangrove-specific indices proposed by Winarso et al. [27]. The index computed from L8 Band 5 (NIR) and Band 6 (SWIR1) was successfully used to map the mangrove forests in the east of Indonesia. The authors reported the efficiency of MI utilizing L8 near- and short-wave infrared bands in their case study; however, the accuracy of mangrove mapping using MI on L8 in different mangrove habitats has not been examined.

Shi et al. [28] suggested an index, i.e., Normalized Difference Mangrove Index (NDMI), to separate mangroves from terrestrial vegetation using L8 Green and SWIR2 wavelengths. NDMI combined with two indices (Spectral Match Degree (SMD); SWIR Absorption Depth (SIAD)) significantly improved mangrove forest identification on L8 in a national nature reserve in the south of China. Additionally, Ali and Nayyar [7] found that NDMI had the highest user accuracy (UA) of mangrove forests among different indices (i.e., NDVI, SAVI, EVI, five MSIs) on L8 in the south of Pakistan, although its producer's accuracy (PA) was not acceptable.

Subtraction of NDVI and NDWI obtained from L8 was considered a new mangrove discriminating index, called Combined Mangrove Recognition Index (CMRI) [25]. The index employs spectral signatures and morphological characteristics to distinguish mangroves from non-mangrove vegetation types. The mangroves of three habitats in India and Bangladesh were well-characterized by CMRI on L8 images. Comparison of CMRI and NDMI showed that the first index could extract L8 pixels covered by mangroves, while the pixels extracted by the second index were a mixture of mangroves and barren land [7].

One of the outstanding characteristics of mangrove ecosystems is periodic submerging of moderate and low-height intertidal stands by fluctuating tide levels. Consequently, the spectral signature of mangroves can be influenced by submerging or emerging status of canopy. Mapping mangrove forests with single satellite images, without consideration of tide levels, may result in under- or overestimation of their extent. A recent study by Xia et al. [41] proposed an index, the Submerged Mangrove Recognition Index (SMRI), using two satellite images of high- and low-tide levels to accurately identify mangrove forests. The SMRI considers NIR wavelength and NDVI to discriminate submerged mangroves that are not characterized by object-oriented classification of NDVI images. The procedure significantly enhances the classification accuracy of mangrove forests. A number of researchers, namely Li et al. [42] and Xia et al. [26], utilized the SMRI as a reliable analytical instrument to map mangrove forests on L8 data.

Mapping mangrove forests within a nature reserve in the south of China, Wang et al. [43] created a new index (Mangrove Discrimination Index (MDI)) to discriminate mangroves from non-mangrove vegetation on different satellite images. The authors suggested using SWIR1 and SWIR2 wavelengths to compute MDI1 and MDI2, respectively, although their investigation revealed that MDI2 performed better than MDI1 on L8 data. However, Mukhtar et al. [44] found that MDI1 was more efficient than MDI2 for mapping mangrove forests in Indonesia on L8 images.

The Modular Mangrove Recognition Index (MMRI) was proposed by Diniz et al. [45] to enhance the contrast of mangrove forests against non-mangroves in the east of Brazil. The index was a modified version of the Normalized Difference Drought Index (NDDI) (Equation (1)). The NDWI was replaced by the Modified Normalized Difference Water Index (MNDWI) (Equation (2)) [46] to construct MMRI.

$$\text{NDDI} = (\text{NDVI} - \text{NDWI})/(\text{NDVI} + \text{NDWI}), \tag{1}$$

$$\text{MNDWI} = (\text{Green} - \text{SWIR1})/(\text{Green} + \text{SWIR1}), \tag{2}$$

The assessment of MMRI on L8 imagery showed that the index achieved more robust results than NDVI, NDWI, and CMRI in discrimination of mangroves from other vegetation types [45]. Additionally, the index was successfully implemented by Sahadevan et al. [47] to map mangroves in the south of India.

The Mangrove Vegetation Index (MVI) was also constructed with the objective of mangrove mapping on L8 and Sentinel-2 data [22]. The index uses Green, NIR, and SWIR1 wavelengths to rapidly and accurately map mangroves. MVI had successfully discriminated mangroves from non-mangroves within six study sites in the Philippines; however, its performance must still be explored within other habitats and be compared to diverse MSIs available for mangrove mapping.

Recently, a new MSI was developed by Ali and Nayyar [7] to specifically separate mangrove forests from other vegetation types on L8 imagery. The Landsat 8 Mangrove Index (L8MI) uses Advanced Slope-based Spectral Transformation (ASST) (Equation (3)) to maximize the difference between mangroves and other features. The SWIR1 and SWIR2 wavelengths (L8 bands 6 and 7, respectively) (Table A2) can be utilized in Equation (3) to produce ASST_1 and ASST_2, respectively.

$$\text{ASST} = (\text{Deep Blue} - \text{SWIR})/(\text{Deep Blue} + \text{SWIR}), \tag{3}$$

The combination of ASST_1 and ASST_2 images and SAVI was called L8MI_1 and L8MI_2, respectively. A comparative investigation revealed that both versions of L8MI were more efficient than VIs (e.g., NDVI, SAVI) and MSIs (i.e., CMRI, NDMI) in discrimination of mangroves in the south of Pakistan, although the robustness of the index has not been evaluated within other mangrove forests.

Table 1 summarizes the common vegetation and mangrove-specific indices investigated in the present study. The VIs were frequently used to map mangroves in previous studies. Meanwhile, the selected MSIs were the indices that have been developed or successfully implemented to recognize mangroves on L8 imagery.

**Table 1.** The commonly used vegetation indices (VIs) and mangrove-specific indices (MSIs) developed for mangrove mapping. All of the indices have been used on Landsat images.

| | Vegetation Indices | Formula | Reference |
|---|---|---|---|
| **VIs** | NDVI (Normalized Difference Vegetation Index) | $(\text{NIR} - \text{Red})/(\text{NIR} + \text{Red})$ | [30] |
| | SAVI (Soil Adjusted Vegetation Index) | $\frac{(\text{NIR}-\text{Red})\ (1+L)}{\text{NIR}+\text{Red}+L}$ | [33] |
| | NDWI (Normalized Difference Water Index) | $(\text{Green} - \text{NIR})/(\text{Green} + \text{NIR})$ | [36] |
| | EVI (Enhanced Vegetation Index) | $2.5 \times \left( \frac{\text{NIR}-\text{Red}}{\text{NIR}+6\times\text{Red}-7.5\times\text{Blue}+1} \right)$ | [39] |

**Table 1.** *Cont.*

| | Vegetation Indices | Formula | Reference |
|---|---|---|---|
| MSIs | MI (Mangrove Index) | $(NIR - SWIR1/NIR \times SWIR1) \times 10{,}000$ | [27] |
| | NDMI (Normalized Difference Mangrove Index) | $SWIR2 - Green/SWIR2 + Green$ | [28] |
| | CMRI (Combined Mangrove Recognition Index) | $NDVI - NDWI$ | [25] |
| | SMRI (Submerged Mangrove Recognition Index) | $(NDVI_L - NDVI_H) \times \frac{NIR_L - NIR_H}{NIR_H}$ | [41] |
| | MDI (Mangrove Discrimination Index) | $(NIR - SWIR)/SWIR$ | [43] |
| | MMRI (Modular Mangrove Recognition Index) | $(|MNDWI| - |NDVI|)/(|MNDWI| + |NDVI|)$ | [45] |
| | MVI (Mangrove Vegetation Index) | $NIR - Green/SWIR1 - Green$ | [22] |
| | L8MI (Landsat 8 Mangrove Index) | $[ASST > T]$ and $[SAVI > T]$ | [7] |

## *2.4. Classification Algorithm*

The first phase of this study presented a comprehensive comparison of VIS and MSIs (Table 1) to determine which index is the most viable to be implemented in mangrove mapping on L8 imagery. Aside from similar remotely sensed data used within three study sites (i.e., S2, S6, S9), a similar classification algorithm was also used to effectively evaluate the robustness of the indices under the same conditions. The accuracy of different classification algorithms varies among different investigations and it seems difficult to determine the most efficient algorithm in mangrove vegetation mapping. Among diverse classification algorithms that have been used in mangrove mapping on Landsat imagery, the support vector machine (SVM) has shown accurate results. The SVM is a machine learning algorithm that works on statistical nonparametric theory, and the application of linear and nonlinear kernels increases its flexibility in generating decision boundaries. The SVM has been widely used by different authors to identify and map mangroves on remotely sensed data [5,28,41,42].

Baloloy et al. (2020) suggested threshold filtering to separate mangroves from other land covers on MVI images; however, threshold selection may significantly affect the accuracy of mangrove mapping. Therefore, an efficient machine learning algorithm, i.e., SVM, was considered for effective separation of mangroves from other classes (soil, water) in this study that are similarly used to the outputs of all indices. The images of MSIs and VIs were separately used as input to SVM. There are three parameters, i.e., kernel function, penalty, and gamma, which have to be optimized in the classification process by the SVM. Previous research on application of the algorithm in mangrove mapping and preliminary experimental results in the present study have indicated the robustness of the radial basis function (RBF). Therefore, the RBF was considered in this study, in addition to the kernel penalty and gamma values of 100 and 0.1, respectively [48–50]. The procedure can reveal the capability of each index in delineation of mangrove forests on Landsat data.

## *2.5. Accuracy Assessment*

As the last part of phase 1, the performance and capability of the VIs and MSIs (Table 1) were quantitatively compared for depicting mangrove and non-mangrove areas. For such quantitative comparison, approximately 24,000 validation pixels were randomly selected for three land cover types (i.e., 6000 pixels for mangroves, 9000 pixels for soil, 9000 pixels for water) in three study sites: S2, S6, and S9. Regarding the difficulty in obtaining the ground control points of land cover types over such a large extent, images with very high spatial resolution obtained from airborne remote sensing (i.e., aerial photographs of 1992 and 2006 with scale of 1:20,000 and 1:40,000, respectively; aerial images taken by UltraCam-D and Xp digital airborne cameras from 2014 to 2018, with approximate pixel size of 7 cm) and Google Earth historical image data (from 1990 to 2020) were considered as the references data in the present study. The number of random pixels was selected according to approximate estimation of area ratio of each land cover type; 70% of these pixels were considered for training the classifier, and 30% were used for accuracy verification. By constructing the confusion matrix and calculating the accuracy, precision, recall, and F1-score of mangroves,

in addition to overall accuracy (OA) and kappa (K), an accuracy assessment was performed to choose the most robust index within three study sites.

In phase 2, the selected index was implemented to map mangrove forests within all study sites (Figure 1), for the three 10-year periods (Figure 2). Total validation pixels considered for training and test in all study sites were approximately 65,000 random pixels, including 17,000 pixels of mangroves, 23,000 pixels of soil, and 25,000 pixels of water. The accuracy of final maps was ascertained by calculating the accuracy statistics (i.e., accuracy, precision, recall, F1-score), based on a confusion matrix constructed from 30% of total validation pixels in all study sites. The results of the statistics for mangrove class were considered to reflect the accuracy of mangrove mapping on Landsat images.

*2.6. Landscape Metrics*

The spatiotemporal changes of mangrove forests in the south of Iran were monitored by landscape metrics, in addition to total area estimation. The integration of spatial analysis of temporal changes allows a better understanding of the mangrove forest dynamics. The assembly of the metrics considered in this study was composed of six metrics concentrated on evaluation of the shape, quantity, and spatial distribution of the patches of mangrove forests over time.

The number of patches (NP), patch density (PD), mean patch area (AREA_MN), and mean perimeter-area ratio (PARA_MN) were used as a measure of fragmentation of mangrove patches in each study site. With regard to the importance of patch shape, the mean shape index (SHAPE_MN) was implemented, and the connectivity between mangrove patches was evaluated by the patch cohesion index (COH). The landscape metrics of NP, PD, and AREA_MN are related to size of mangrove patches and can be considered as a reliable measure of fragmentation. Two metrics, PARA_MN and SHAPE_MN, are indicators of shape complexity of the mangrove patches. As the value of PARA_MN increases, it shows higher complexity of the patch shapes. When the shapes of patches are close to square in raster datasets, the value of SHAPE_MN is equal to 1; when the shapes become more irregular, the values of the index increase with no limits. The connectivity between mangrove patches of the same landscape can be assessed by COH. The values of COH range from 0 to 100, and the higher the value of the index is, the more aggregated the patches are. The landscape metrics can provide additional information about the spatiotemporal changes of mangroves, such as aggregation of patches and mangrove fragmentation in the study period (i.e., 1990–2020) [51–53].

## 3. Results

### 3.1. Comparison of VIs and MSIs

The results of the accuracy assessment of the VIs and MSIs from the low-tide and multi-tidal Landsat images at three study sites (S2, S6, and S9) selected for the first phase are presented in Table 2. As the main aim of this study was mangrove mapping, the accuracy of mangrove delineation (i.e., statistics of accuracy, precision, recall, and F1-score) was reported in addition to OA and K. Mangrove detection in the selected study sites using 14 VIs and MSIs showed contrasting results. In S2, for instance, the accuracy and F1-score of mangrove mapping on the SAVI image (0.86 and 0.53, respectively) were less than the MVI results (0.95 and 0.86, respectively); however, the accuracy and F1-score of mangrove delineation on the SAVI image (0.93 and 0.91, respectively) were higher than the MVI results (0.90 and 0.87, respectively) in S6 (Table 2). Mangrove mapping on the NDVI images, as the commonly used index for mangrove mapping in the literature, showed lower accuracy and F1-scores compared to some VIs and MSIs in the selected study sites (e.g., 0.77 and 0.53 in S2). In general, the accuracy and F1-score of mangrove mapping were the highest on the SMRI images in S2, S6, and S9.

**Table 2.** The accuracy assessment of mangrove forest mapping on the images of VIs and MSIs determined from Landsat 8 data classified by SVM algorithm in three study sites, S2, S6, and S9, shown in Figure 1 (accuracy, precision, recall and F1-score of mangroves; OA: overall accuracy; K: kappa coefficient).

| Site | Criteria | VIs | | | | | MSIs | | | | | | | | |
|---|---|---|---|---|---|---|---|---|---|---|---|---|---|---|---|
| | | NDVI | SAVI | NDWI | EVI | MI | NDMI | CMRI | SMRI | MDI-1 | MDI-2 | MMRI | MVI | L8MI-1 | L8MI-2 |
| S2 | Accuracy | 0.77 | 0.86 | 0.87 | 0.87 | 0.58 | 0.95 | 0.89 | 0.95 | 0.45 | 0.41 | 0.67 | 0.95 | 0.95 | 0.94 |
| | Precision | 0.62 | 0.40 | 0.38 | 0.46 | 0.95 | 0.70 | 0.48 | 0.98 | 0.86 | 0.74 | 0.73 | 0.78 | 0.70 | 0.62 |
| | Recall | 0.45 | 0.82 | 0.94 | 0.83 | 0.32 | 0.99 | 0.96 | 0.82 | 0.25 | 0.22 | 0.35 | 0.96 | 0.99 | 0.99 |
| | F1-score | 0.53 | 0.53 | 0.55 | 0.6 | 0.48 | 0.82 | 0.64 | 0.89 | 0.39 | 0.34 | 0.48 | 0.86 | 0.82 | 0.77 |
| | OA (%) | 77.4 | 86.1 | 87.1 | 87.3 | 58.8 | 95.9 | 89.2 | 95.4 | 41.9 | 45.9 | 67.7 | 95.1 | 94.8 | 95.9 |
| | K | 0.64 | 0.77 | 0.78 | 0.79 | 0.40 | 0.80 | 0.82 | 0.94 | 0.12 | 0.18 | 0.48 | 0.90 | 0.74 | 0.80 |
| S6 | Accuracy | 0.87 | 0.93 | 0.89 | 0.94 | 0.89 | 0.94 | 0.90 | 0.97 | 0.88 | 0.65 | 0.67 | 0.90 | 0.97 | 0.95 |
| | Precision | 0.97 | 0.96 | 0.94 | 0.96 | 0.90 | 0.98 | 0.96 | 0.93 | 0.92 | 0.91 | 0.97 | 0.95 | 0.93 | 0.85 |
| | Recall | 0.74 | 0.87 | 0.79 | 0.91 | 0.83 | 0.88 | 0.81 | 0.99 | 0.87 | 0.50 | 0.52 | 0.80 | 0.99 | 0.98 |
| | F1-score | 0.84 | 0.91 | 0.86 | 0.93 | 0.86 | 0.93 | 0.88 | 0.96 | 0.89 | 0.65 | 0.68 | 0.87 | 0.96 | 0.91 |
| | OA (%) | 87.2 | 89.7 | 89.2 | 85.5 | 95.3 | 97.8 | 95.5 | 96.9 | 81.2 | 42.4 | 50.3 | 96.7 | 95.2 | 97.6 |
| | K | 0.81 | 0.90 | 0.83 | 0.83 | 0.89 | 0.90 | 0.91 | 0.97 | 0.69 | 0.23 | 0.31 | 0.90 | 0.91 | 0.91 |
| S9 | Accuracy | 0.93 | 0.95 | 0.90 | 0.95 | 0.76 | 0.95 | 0.95 | 0.96 | 0.66 | 0.57 | 0.78 | 0.95 | 0.95 | 0.95 |
| | Precision | 0.95 | 0.94 | 0.91 | 0.94 | 0.69 | 0.88 | 0.98 | 0.98 | 0.87 | 0.71 | 0.84 | 0.97 | 0.86 | 0.87 |
| | Recall | 0.80 | 0.86 | 0.72 | 0.87 | 0.48 | 0.96 | 0.85 | 0.87 | 0.39 | 0.30 | 0.51 | 0.83 | 0.96 | 0.96 |
| | F1-score | 0.87 | 0.90 | 0.81 | 0.91 | 0.57 | 0.92 | 0.91 | 0.92 | 0.54 | 0.42 | 0.63 | 0.90 | 0.91 | 0.91 |
| | OA (%) | 91.7 | 85.5 | 80.4 | 84.1 | 76.5 | 98.4 | 90.3 | 94.2 | 66.2 | 57.6 | 73.9 | 89.5 | 97.2 | 96.7 |
| | K | 0.90 | 0.83 | 0.74 | 0.73 | 0.65 | 0.91 | 0.85 | 0.95 | 0.49 | 0.36 | 0.60 | 0.84 | 0.92 | 0.90 |

In general, the results indicated that the detection of mangrove forests through the SVM classification of the SMRI images was the most accurate approach for the selected study sites (i.e., S2, S6, S9). In addition to the quantitative assessment of mangrove delineation using VIs and MSIs (Table 2), the results of all 14 indices in S6 (Hara Roud-e Gaz protected area in Figure 1) were presented in Figure 3, as an example for visual interpretation. The identification of mangrove patches was not similar on the images of different indices. Mangroves were overestimated by some indices (e.g., MDI-1, MDI-2), while a few indices such as L8MI-1 and L8MI-2 underestimated the mangroves.

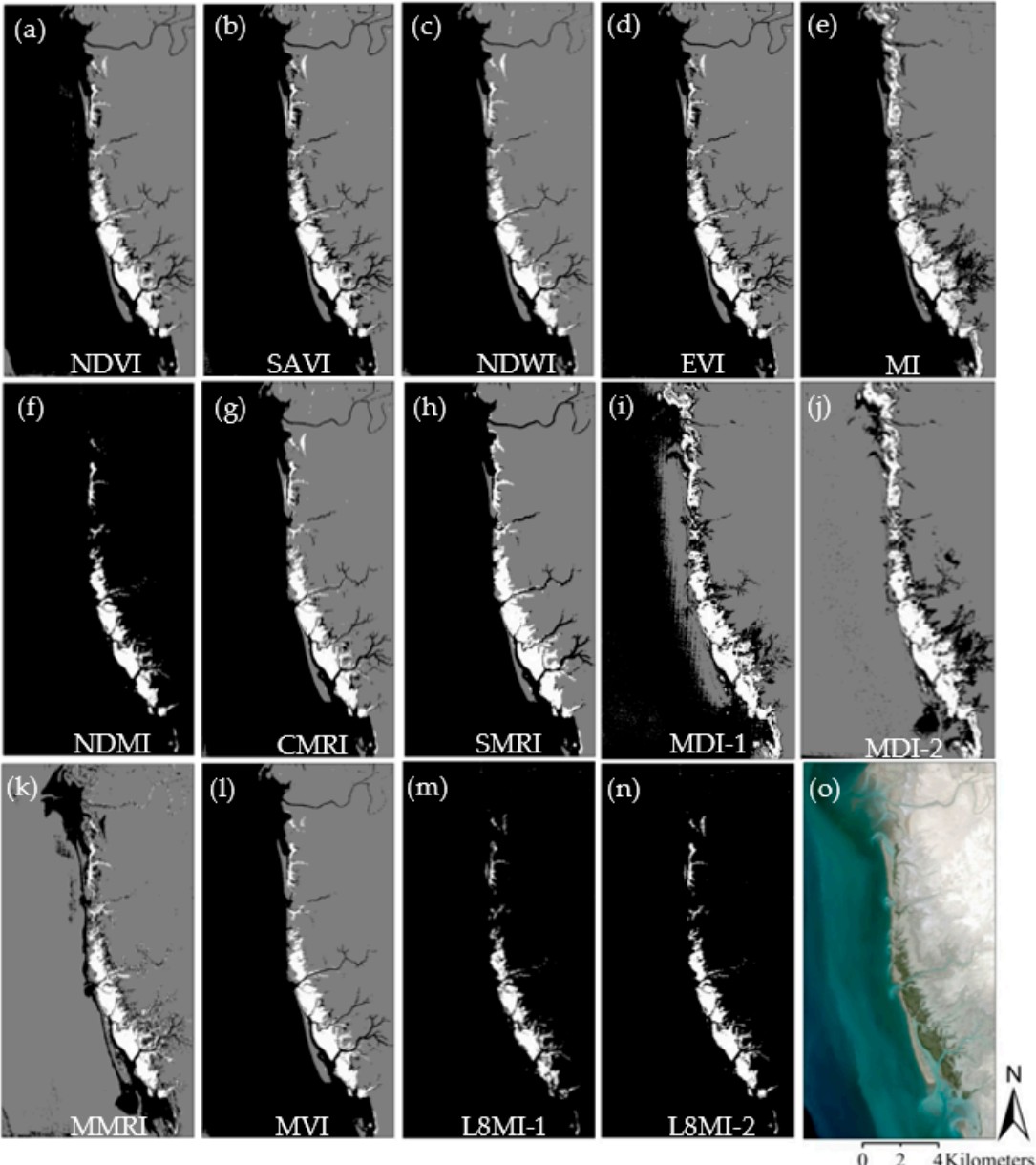

**Figure 3.** Results of the vegetation indices (i.e., NDVI, SAVI, NDWI, and EVI, shown in panels (**a**–**d**)) and mangrove-specific indices (i.e., MI, NDMI, CMRI, SMRI, MDI, MMRI, MVI, and L8MI, shown in panels (**e**–**n**)) from the low-tide and multi-tidal Landsat 8 image in S6 (Hara Roud-e Gaz protected area in Figure 1). (**o**) is the low-tide Landsat 8 image in S6. (Note: MDI-1 and L8MI were computed by SWIR1 (MDI-1 and L8MI-1, respectively) and SWIR2 (MDI-2 and L8MI-2, respectively) wavelengths. White areas represent mangrove forests).

Additionally, the mangrove boundaries delineated by the SMRI index were overlaid on L8 imagery (Figure 4a–c) and high resolution true-color Google Earth images (Figure 4d–f) in three study sites: S2, S6, and S9. The visual interpretation of the SMRI and NDVI results showed that SMRI perfectly mapped both dense and sparse mangrove patches; however, NDVI detected only dense mangroves.

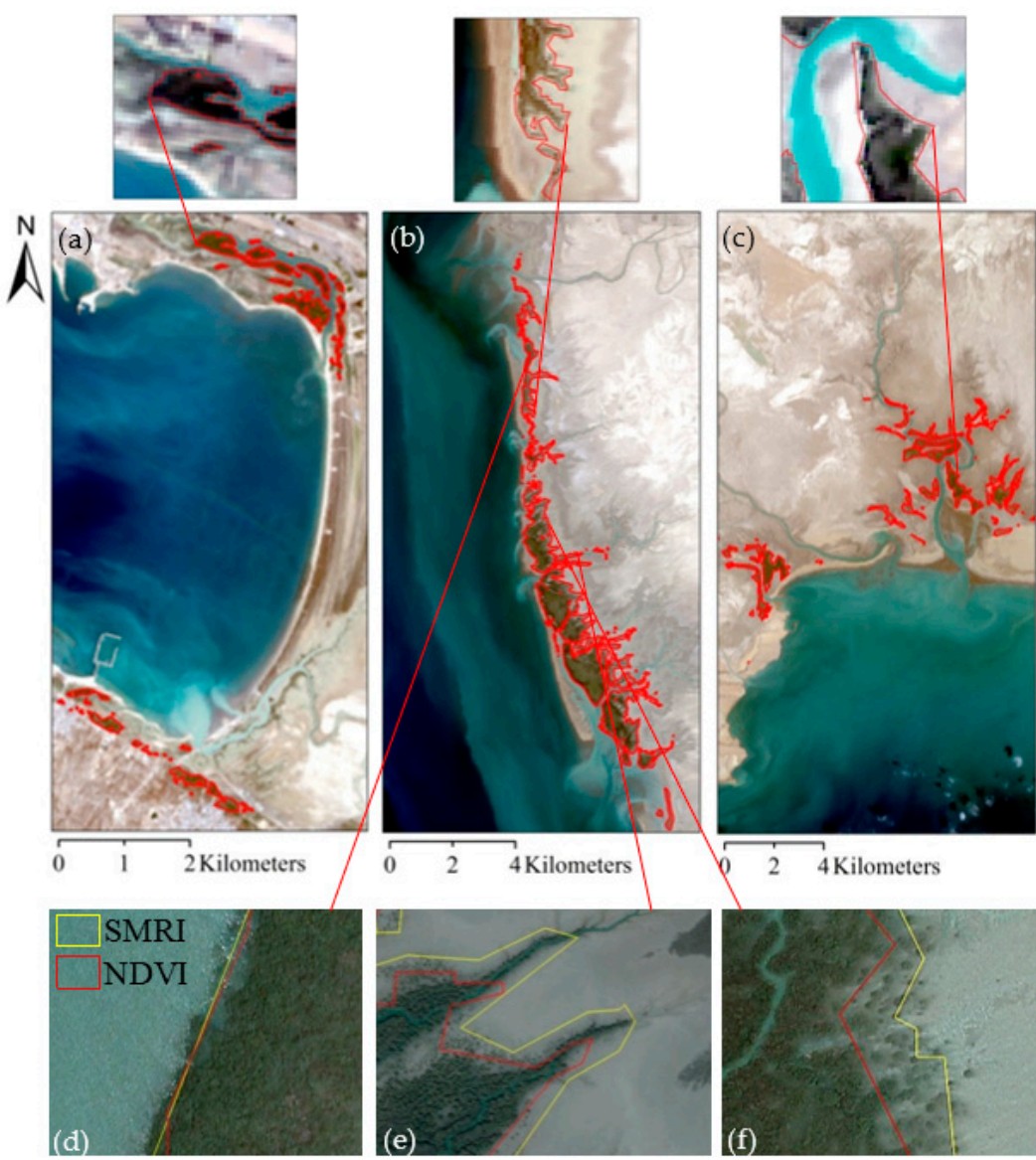

**Figure 4.** Classification results of mangrove forests in S2 (Nayband National Park) (**a**), S6 (Hara Roud-e Gaz protected area) (**b**), and S9 (Bahookalat protected area) (**c**) from SMRI images. Panels (**a**–**c**) are true color composites of the low-tide Landsat 8 imagery (image date: 8 March 2020). Panels (**d**–**f**) compare the results of mangrove mapping on the SMRI and NDVI images on very high-resolution Google Earth images.

The quantitative and qualitative evaluation of the VIs and MSIs showed that the SMRI images were the most reliable data for mangrove mapping within the selected study sites. The SMRI index was, therefore, used to detect mangroves within all the study sites in the study period. Table 3 shows the accuracy assessment of the mangrove maps within all the study sites (Figure 1) in the study years obtained from the SMRI images of the sites classified by SVM algorithm. It should be noted that there were no mangrove forests in S8 before 2010, so the mangroves were established after 2010. Therefore, no accuracy assessment was provided in 1990 and 2000 within S8.

**Table 3.** The accuracy assessment of mangrove forest mapping from 1990 to 2020 with 10-year lags on the SMRI images of all study sites (Figure 1), classified by SVM algorithm (accuracy, precision, recall, and F1-score of mangroves; OA: overall accuracy; K: kappa coefficient).

| Sites | 1990 | | | | | | 2000 | | | | | |
|---|---|---|---|---|---|---|---|---|---|---|---|---|
| | Accuracy | Precision | Recall | F1-Score | OA (%) | K | Accuracy | Precision | Recall | F1-Score | OA (%) | K |
| S1 | 0.91 | 0.86 | 0.74 | 0.80 | 91.8 | 0.87 | 0.88 | 0.90 | 0.64 | 0.75 | 88.7 | 0.82 |
| S2 | 0.90 | 0.93 | 0.70 | 0.80 | 90.1 | 0.84 | 0.74 | 0.70 | 0.53 | 0.60 | 91.1 | 0.86 |
| S3 and S4 | 0.94 | 0.97 | 0.88 | 0.92 | 94.4 | 0.91 | 0.94 | 0.96 | 0.89 | 0.92 | 94.8 | 0.92 |
| S5 | 0.94 | 0.95 | 0.87 | 0.91 | 94.3 | 0.91 | 0.94 | 0.96 | 0.86 | 0.91 | 93.9 | 0.91 |
| S6 | 0.94 | 0.95 | 0.88 | 0.91 | 94.2 | 0.91 | 0.94 | 0.95 | 0.89 | 0.92 | 94.8 | 0.92 |
| S7 | 0.92 | 0.95 | 0.77 | 0.85 | 93.1 | 0.89 | 0.93 | 0.97 | 0.76 | 0.86 | 93.5 | 0.89 |
| S8 | - | - | - | - | - | - | - | - | - | - | - | - |
| S9 | 0.95 | 0.94 | 0.89 | 0.92 | 95.7 | 0.93 | 0.96 | 0.95 | 0.90 | 0.92 | 96.1 | 0.94 |

| Sites | 2010 | | | | | | 2020 | | | | | |
|---|---|---|---|---|---|---|---|---|---|---|---|---|
| | Accuracy | Precision | Recall | F1-Score | OA (%) | K | Accuracy | Precision | Recall | F1-Score | OA (%) | K |
| S1 | 0.85 | 0.83 | 0.58 | 0.68 | 85.6 | 0.77 | 0.89 | 0.86 | 0.66 | 0.75 | 89.4 | 0.83 |
| S2 | 0.91 | 0.90 | 0.76 | 0.82 | 91.8 | 0.87 | 0.95 | 0.98 | 0.82 | 0.89 | 95.4 | 0.94 |
| S3 and S4 | 0.92 | 0.99 | 0.83 | 0.90 | 92.7 | 0.89 | 0.95 | 0.96 | 0.92 | 0.94 | 95.8 | 0.93 |
| S5 | 0.94 | 0.96 | 0.87 | 0.91 | 94.3 | 0.91 | 0.95 | 0.96 | 0.90 | 0.92 | 95.2 | 0.92 |
| S6 | 0.94 | 0.98 | 0.87 | 0.92 | 94.5 | 0.91 | 0.97 | 0.93 | 0.99 | 0.96 | 96.9 | 0.97 |
| S7 | 0.96 | 0.98 | 0.84 | 0.90 | 96.2 | 0.93 | 0.93 | 0.98 | 0.76 | 0.85 | 93.5 | 0.89 |
| S8 | 0.88 | 0.90 | 0.59 | 0.71 | 88.4 | 0.81 | 0.97 | 0.98 | 0.87 | 0.92 | 97.5 | 0.95 |
| S9 | 0.94 | 0.98 | 0.84 | 0.91 | 94.9 | 0.92 | 0.96 | 0.98 | 0.87 | 0.92 | 94.2 | 0.95 |

### 3.2. Mangrove Area Change Assessment

The SMRI-derived mangrove area within each study site was compared from 1990 to 2020 (Figure 5). The results indicated dissimilarity in area changes of mangroves between the study sites. An area increase over the mangrove habitats was observed in S1, S2, and S6 to S9 during the period 1990–2020. The highest increase rate was observed in S2 and S8, auxh that the total area of mangroves changed from 64.9 and 0.0 ha in 1990 to 243.9 and 99.6 ha in 2020, respectively (Figure 5b,h). No mangrove forests were observed in S8 on the SMRI images in 1990 and 2000, although small patches were established in 2010 (20.5 ha) and rapidly developed in 2020 (99.6 ha) (Figure 5h). Alternatively, the study sites of S3 and S4, with the largest mangrove distribution in 1990 (6364.1 and 1815.3 ha, respectively), showed no significant increase in 2020 (6502.6 and 2041.6 ha, respectively). However, a sharp decline was observed in S3 and S4 from 2000 (6782.4 and 2129.2 ha, respectively) to 2010 (6338.3 and 1844.2 ha, respectively) (Figure 5c,d). Additionally, the mangrove areas were approximately doubled in S7 and S9 within the study period (from 541.7 and 322.2 ha in 1990 to 1111.2 and 645.9 ha in 2020, respectively) (Figure 5g,i). In general, the distribution of mangrove forests in all study sites increased from 10,706.2 to 12,177.3 ha during the decade 1990–2000 and decreased to 11,749.9 in 2010 but expanded to 13,019.1 ha in 2020.

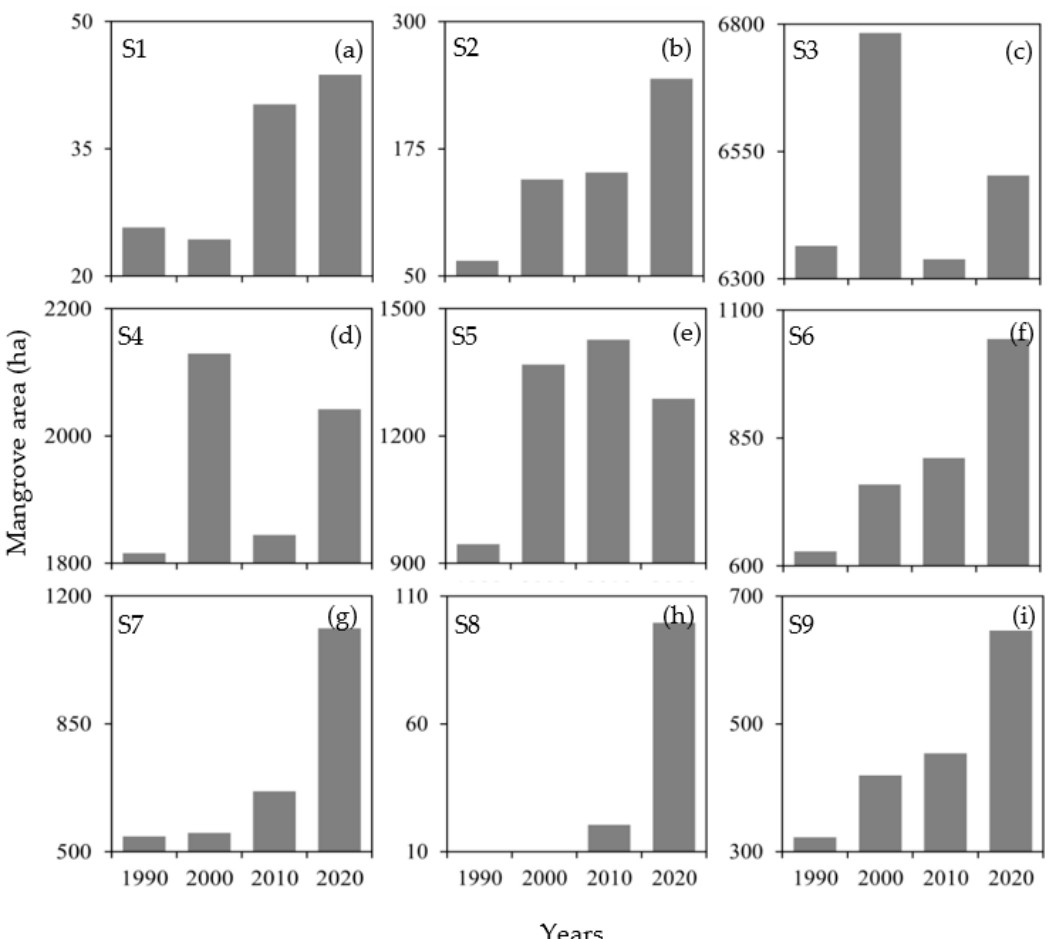

**Figure 5.** Mangrove forest dynamics over time (from 1990 to 2020) within each mangrove site (S1–S9 in Figure 1) on SMRI images presented in separated panels (**a**–**i**).

Figure 5 shows the cumulative changes of mangrove coverage during the period 1990–2020 within each study site; however, the rates of area increase and decrease in each 10-year period are also important for the global assessment of mangrove forests. Figure 6 indicates the increase and decrease percentages of mangrove areas in the three decades of 1990–2000, 2000–2010, and 2010–2020, within each study site. A closer look at Figure 6

shows that the mangroves in S2 (54.6%, Figure 6b) and S8 (77.8%, Figure 6g) had the highest increase rate in the temporal interval from 2010 to 2020, whereas the highest decrease was registered in S1 (49.4%, Figure 6a) in the interval from 1990 to 2000. The average rate of area changes in three decades (20.0%, 14.1%, and 33.2% increases in the 1990–2000, 2000–2010, and 2010–2020 epochs, respectively; Figure 6i) suggests an increase in total mangrove coverage in southern Iran. However, it is important to note that the results of mangrove area change assessment were slightly different in some study areas, when the amounts of increase and decrease were considered instead of their relative proportions. In S3 and S4, for instance, the highest increase (483 ha in the decade 1990–2000) and decrease (450 ha in the decade 2000–2010) were observed, although their standardized values (5.4% increase and 5.5% decrease, respectively) were not significant, due to the largest distribution of mangroves in the two study sites.

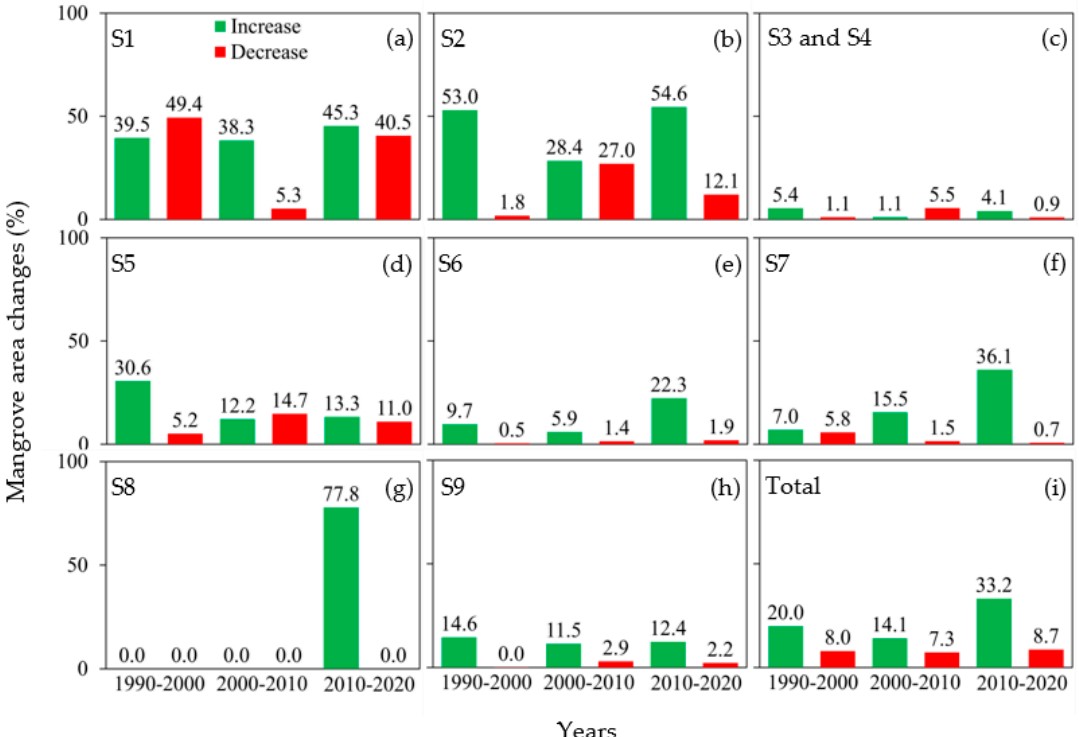

**Figure 6.** Mangrove area increase and decrease over time (three decades of 1990–2000, 2000–2010, and 2010–2020 epochs) in all major mangrove sites (S1–S9 in Figure 1) on SMRI images. Panels (**a**–**h**) show changes of mangroves in each study site, and panel (**i**) exhibits the total changes in all sites.

The total spatiotemporal changes of mangrove forests within the study sites in the temporal interval between 1990 and 2020 are presented in Figure 7. The visual interpretation of spatial changes of mangroves revealed that their expansions were landward in some study sites (S6, S7), although their development was seaward in S5 and S9. The mangrove losses mostly occurred landward, which are well shown in two subsites of S5.

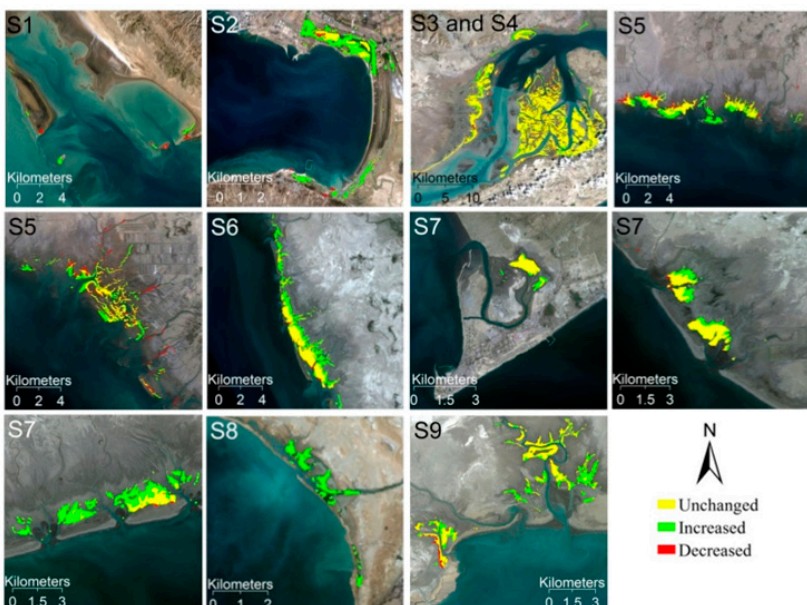

**Figure 7.** Spatiotemporal changes of mangrove forests in all major mangrove sites (S1–S9 in Figure 1) on SMRI images between 1990 and 2020 in southern Iran.

### 3.3. Evaluation of Landscape Metrics

Further analyses using the landscape metrics were carried out to monitor the spatial distribution changes of mangrove patches within the study sites from 1990 to 2020 (Figure 8). The NP and PD were significantly increased in S2, S6, S7, S8, and S9; however, the values of the metrics decreased in S1, S3–S4, and S5 during the study period. The highest increasing rate was observed from 1990 to 2000 in S2 and S6 and from 2010 to 2020 in S7, while the metrics gradually increased during the study period in S8 and S9. Additionally, the decrease in the metrics was sharp from 1990 to 2000 in S1 and S5, although the metrics suddenly decreased from 2000 to 2010 in S3–S4 (Figure 8a–h). It seems that establishment and development of mangrove patches caused an increasing trend of the NP and PD in a number of study sites (i.e., S2, S6, S7, S8, S9).

The AREA_MN showed an increasing tendency from 1990 to 2020 in all study sites, although the tendency was not sharp in S5 and S6 (Figure 8i–l). During the last epoch (i.e., 2010–2020), the metric slightly decreased within S1, S3–S4, and S7, while an increasing trend was observed from 1990 to 2010. Moreover, the metric gradually increased within the other study sites. The AREA_MN metric revealed that the size of the mangrove patches increased in all study sites, although their size became smaller within three sites in the last decade.

The values of PARA_MN showed a complexity reduction from 1990 to 2020 in S1, S5, S6, and S7, while a contrasting trend was found in other study sites. The highest rise of complexity was observed in S9, whereas S1 experienced the greatest reduction (Figure 8m–p).

The SHAPE_MN for all study sites was greater than 1 (except S8 in 1990 and 2000), indicating that the average patch shape in all study sites was noncircular. Values between 1.1 and 1.7 revealed a rectangular shape of the patches (Figure 8q–t). Additionally, the increasing trend of SHAPE_MN in all study sites showed the greater complexity of mangrove patches in 2020 compared to 1990.

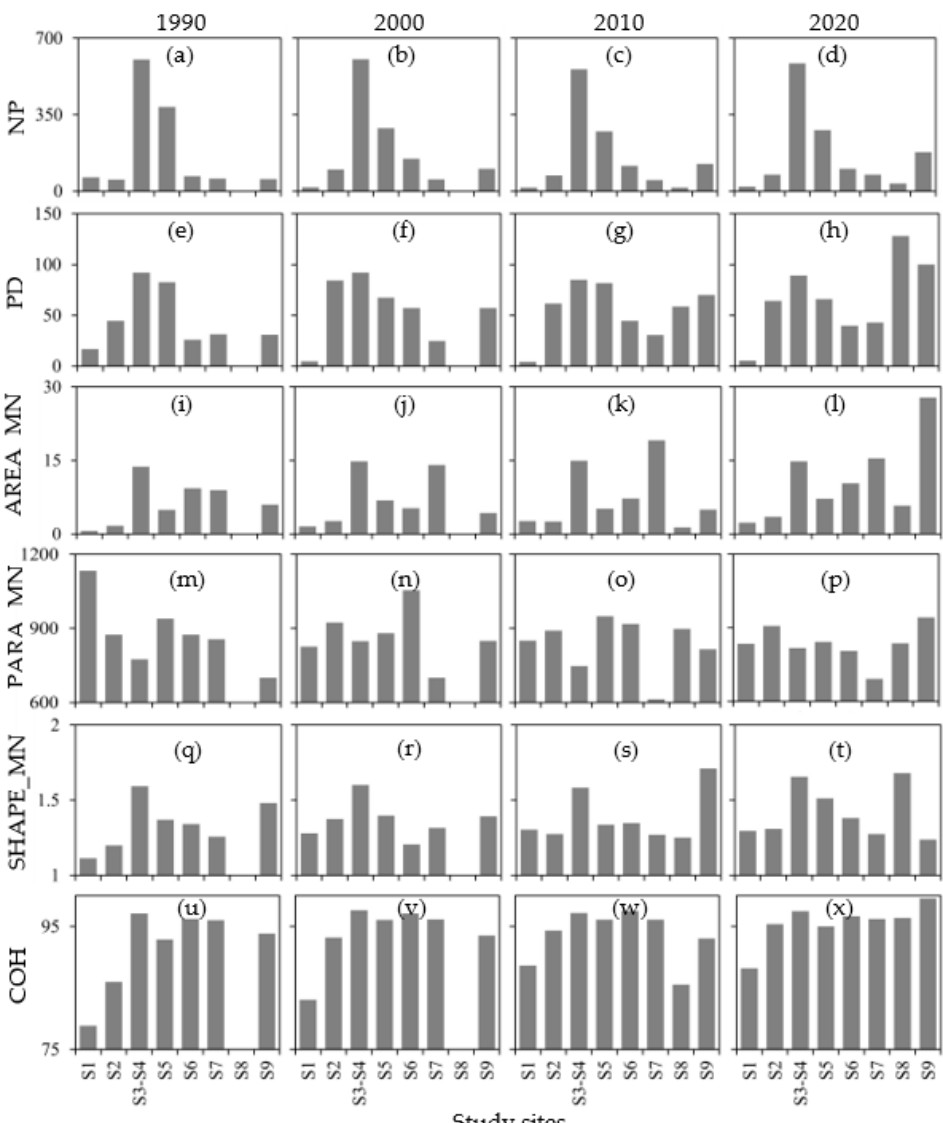

**Figure 8.** Changes of the landscape metrics of mangrove patches within study sites of S1–S9 (Figure 1) presented in panels (**a**–**x**), respectively, in the study period (1990, 2000, 2010, 2020). The metrics are NP (number of patches), PD (patch density), AREA_MN (mean patch area), PARA_MN (mean perimeter-area ratio), SHAPE_MN (mean shape index), and COH (patch cohesion index).

The variation in the COH showed an increasing tendency within S1, S2, S7, S8, and S9, indicating a stronger connectivity of mangrove patches in the sites in 2020 compared to 1990. The tendency was also increasing in S5 and S6 during 1990–2010; however, a sharp decrease was observed in both sites from 2010 to 2020. There was no significant change in the metric values in S3–S4 during the study period (Figure 8u–x).

## 4. Discussion

### 4.1. Robustness of VIs and MSIs in Mangrove Forest Mapping

Mangrove forests are located along intertidal zones in relatively small patches; therefore, a lack of considering the high and low tidal conditions would result in the misclassification between mangroves and water. Application of VIs not specifically designed for mangrove identification may cause a mixture of pixels covered by mangrove forests and water. To solve the problem, a number of MSIs have been proposed in the literature; however, their efficiency in the study sites of the present research was not clear. To accurately map and monitor the mangrove forests along the southern coast of Iran in this study, we,

therefore, compared the accuracy of mostly cited VIs and almost all MSIs that have been developed for mangrove mapping on Landsat data (Table 1). Efficiency of the VIs and MSIs on Landsat imagery was evaluated within three study sites (i.e., S2, S6, S9) (Figure 1) in the first phase. In general, our findings in the selected study sites (Table 2) indicated less effectiveness of the VIs (Table 1), such as NDVI, as the commonly used index for mangrove mapping [12,19,24].

The results of accuracy statistics such as the accuracy and F1-score of mangroves indicated the robustness of most of the MSIs (Table 2), although the maps obtained from the SMRI images showed the highest accuracy. The visual interpretation of the mangrove maps (Figure 4) and the values of the accuracy statistics (Table 2) suggested that SMRI was the most appropriate index within the study sites. The result supports the findings of previous studies, that SMRI provides great opportunities in the efficient discrimination of mangrove forest patches on Landsat imagery [26,41,42]. To the best of our knowledge, a comprehensive comparison of the performance of VIs and MSIs was not considered in the literature, and a limited number of indices were evaluated in studies that proposed a new mangrove index (Table 4). Our findings imply that capability of VIs and MSIs to map mangroves on Landsat data may be inconsistent in different mangrove forests. Diniz et al. [45], for instance, reported that MMRI was more efficient than CMRI in mapping Brazil's mangrove forests, while CMRI showed better results than MMRI in our study sites (Table 2). Future studies may explore the influence of the biophysical characteristics of mangrove forests (e.g., canopy cover, intertidal conditions) on the performance of vegetation and mangrove indices.

**Table 4.** A summary of the vegetation indices proposed and compared by various authors to map mangrove forests on Landsat 8 images.

| Reference | Compared Vegetation Indices | The Proposed Index | The Selected Index |
|---|---|---|---|
| Gupta et al. [25] | SR, NDVI, NDWI, SAVI, CMRI | CMRI | CMRI |
| Wang et al. [43] | SR, DVI, NDVI, EVI, MDI | MDI | MDI |
| Diniz et al. [45] | NDVI, NDWI, CMRI, MMRI | MMRI | MMRI |
| Ali and Nayyar [7] | RVI, EVI, NDVI, SAVI, CMRI, NDMI, L8MI | L8MI | L8MI |
| Xia et al. [26] | RVI, EVI, NDVI, SAVI, MRI, SMRI | SMRI | SMRI |

SR: Simple Ratio, RVI: Ratio Vegetation Index, DVI: Difference Vegetation Index, MRI: Mangrove Recognition Index.

Baloloy et al. [22] explained that the difference in the reflectance of mangrove and terrestrial vegetation canopies is greater in the SWIR1 wavelength compared to SWIR2. However, Ali and Nayyar [7] found no significant difference between the reflectance of mangrove and terrestrial vegetation in the SWIR1 and SWIR2 wavelengths and suggested to use both wavelengths in their proposed MSI (i.e., L8MI). The results of the present study were in accordance with the achievements of Baloloy et al. [22]. Table 2 showed that the accuracy of mangroves delineated on MDI-1 and L8MI-1 images (calculated by SWIR1) was greater than the accuracy of mangroves mapped on MDI-2 and L8MI-2 images (based on SWIR2). In comparison with SWIR2, reflectance of the SWIR1 wavelength is significantly lower for mangroves [22,43] and, thus, can be considered as the input wavelength (Band 6 in L8) for the index-based mapping of mangrove forests on Landsat data.

### 4.2. Long-Term Changes of Iran's Mangrove Forests

The present study can be considered as the first comprehensive assessment of Iran's mangrove forests, which systematically mapped and monitored the extent of mangrove patches distributed along the coast of southern Iran using a robust MSI. The long-term increasing rate of mangrove forest area found in the present study is in accordance with the achievements of previous investigations along the southern coasts of Iran [14] and case studies in major sites [54,55]. The latest estimate of Iran's mangrove forests is for the year 2017, with a total area of 9403 ha, using NDVI images derived from Landsat

data [14]. Moreover, the study reported the total area of mangrove forests in 1977 (4735 ha), 1989 (6052 ha), and 2000 (8015 ha). However, the estimated total areas were significantly different from the amounts obtained in the present study (i.e., 10,706.2 ha in 1990, 12,177.3 ha in 2000, 11,749.9 ha in 2010, 13,019.1 ha in 2020). Considering the higher accuracy of SMRI than NDVI in mangrove mapping on Landsat imagery (Figure 4d–f), it is likely that the mangrove-delineation approach is the main reason for the observed differences in the estimation of the total area of mangroves in southern Iran. In a comprehensive assessment in 2011 by Danehkar et al. [56], the national mangrove area was 11,017.5 ha based on analysis of IRS imagery, which is similar to our findings for 2010.

The spatial distribution of mangroves showed distinct trends between the study periods. In the first period, from 1990 to 2000, the trend was upward with almost a 13.7% increase in the total area of mangroves. In line with our findings, Mafi-Gholami et al. [19] also demonstrated that favorable environmental conditions before 1998 resulted in the expansion of mangrove forests in three sites: S3–S4, S5, and S7. This may explain the upward trend observed in the total area of mangrove forests within all sites. The second period, i.e., between 2000 and the 2010 epoch, indicated a downward trend, with approximately a 3.5% loss of mangroves. A similar status was observed by Mafi-Gholami et al. [19] within three study sites, as a result of extensive and continuous drought that started from 1998 in the region. Additionally, Etemadi et al. [20] reported a significant reduction in mangrove area in S2 after 1997, because of severe drought and human-induced impacts such as industrial effluent discharge to estuaries and rapid urban development. Our findings indicated that the loss of mangrove forests during the period 2000–2010 was only observed in S3 and S4 (Figure 5), while other sites experienced an increasing rate. Natural stresses (e.g., climate change) and anthropogenic pressures are probably the main drivers of the reduction in mangrove forests in S3 and S4. It should be noted that S3 and S4 are the most important mangrove sites, with an area of 8544.2 ha, which comprise more than 65.6% of the total area (in 2020). Therefore, the decrease in mangroves in S3 and S4 (6.5% and 13.4%, respectively) from 2000 to 2010 resulted in the downward trend of the total area of mangroves, despite the increasing rates observed in the other sites. In the third period, from 2010 to 2020, the trend was upward, with an increasing rate of 10.8% in the total area of mangroves. The observed increasing rate is mainly due to plantation activities by Natural Resources Administrations (NRAs), in cooperation with local people; growing awareness of public and the local authorities; and more severe conservation status of the forests, in the forms of biosphere reserves, protected areas, and national parks, during the last decade. The efforts made by the NRAs probably caused a significant increasing rate in the mangrove spatial extent between 2010 and 2020 epoch (Figure 6i); otherwise, the impacts of climate change could limit the development of mangrove forests, as predicted by Mafi-Gholami et al. [18]. In S8, for instance, no mangrove forest was observed on Landsat imagery in 2000, while the site was covered by mangroves in 2010 (20.5 ha) and expanded to 99.6 ha in 2020 (Figures 5h and 6g). However, S5 was the only site that experienced landward mangrove deforestation during the period (Figures 5e and 7), likely due to the impacts of urbanization and development of shrimp aquaculture. In general, our findings indicated that the spatial extent of mangrove forests in all study sites increased, with an annual rate of 0.72% during the study period, i.e., 1990–2020. The seaward and landward spatial distributions of the mangroves in most study sites (see Figure 7 for more details) can be considered as the result of natural regeneration, because of the conservation and afforestation by the NRAs.

*4.3. Mangrove Forest Monitoring through Landscape Metrics*

Although remote-sensing based information on changes of mangrove spatial extents allow a deep understanding of the dynamics of mangrove forests in southern Iran, it seems essential to assess the dynamics of mangrove patches throughout robust landscape metrics to obtain reliable interpretations regarding the observed temporal fluctuations in the country's mangrove total areas.

The NP and PD metrics showed a decreasing tendency in some study sites (i.e., S1, S3–4, S5), while an increasing tendency was observed in other study sites (i.e., S2, S6, S7, S8, S9) from 1990 to 2020. It seems that the mangrove patches became more compact in S1, S3–4, and S5, in contrast to other study sites, where the mangrove patches experienced fragmentation. However, AREA_MN and SHAPE_MN indicated an increasing trend during 1990–2020 within all sites, which demonstrates more irregularity of the mangrove patches. Considering PARA_MN, an increasing complexity was observed in most study sites, except S1, S5, S6, and S7. The COH metric revealed a higher connectivity among the mangrove patches from 1990 to 2020, although it was less strong in S3–4, S6, and S7 (Figure 8). In general, the mangrove patches in S1 and S5 became less in number, with more compactness and less complexity, with strong connectivity that can be considered as a degradation of the mangroves from 1990 to 2020. The mangroves in S2, S8, and S9 experienced an increasing number of patches, with irregularity and high complexity, with strong connectivity that shows restoration of the sites during the epoch. In S3 and S4, the number of mangrove patches was reduced, the irregularity and complexity increased, and the connectivity became weak, indicating slight degradation in some parts of the sites. Finally, the decreasing number of mangrove patches in S6 and S7, with less complexity and weak connectivity, may account for the significant degradation of the sites in the study period.

Mangrove forests are mainly influenced by natural and anthropogenic threats. From the perspective of natural threats, temperature change and sea level rise, as the impacts of climate change, are likely to be the most important factors affecting mangroves. Previous works have reported contrasting results on the northern coasts of the Persian Gulf and Gulf of Oman, about the natural factors that significantly threaten world mangrove forests. For instance, Goharnejad et al. [57] found no significant trend in sea level rise from 1990 to 2008 within four tide gauges stations installed by the Iranian Tide Gauge Network along the coasts. However, Al-Subhi and Abdulla [58] explored an increasing linear trend of 2.58 mm in sea level rise per year from 1992 to 2020. Additionally, Al Senafi [59] found an overall loss of heat (cooling) in sea surface temperature (SST) during the 1982–2020 epoch, while Al-Subhi and Abdulla [58] reported SST increase with a linear trend of 0.027 °C per year from 1992 to 2020. The findings of this study have to be seen in light of variations in natural factors that could be accurately addressed at site-level in future investigation. Additionally, unofficial reports have shown that the landward expansion of mangrove spatial distribution in different sites (e.g., S6, S7, S8, S9) is probably due to plantation activities by NRAs and management of mangroves under biosphere reserves, protected areas, and national parks. However, it should be noticed that the development of mangrove forests observed in some sites (e.g., S1, S2) and the degradation explored in other sites (e.g., S3, S4) are mainly derived by anthropogenic factors, considering the achievements of the previous research and the findings of the present study.

## 5. Conclusions

The distribution and spatiotemporal alteration of Iran's mangrove forests along the shorelines of the Persian Gulf and Gulf of Oman during the period of 1990 to 2020 were explored using Landsat time series data and landscape metrics. First, we compared commonly used and recently developed vegetation indices on L8 data, based on our hypotheses. The detection of mangrove patches within three sites was carried out through four VIs and eight MSIs. The SMRI was validated as the most effective index (F1-score $\geq 0.89$) for mangrove identification in the selected sites. Later, we implemented the SMRI on the Landsat time series data, within nine sites, to obtain mangrove areas and spatial distributions for 1990, 2000, 2010, and 2020. The SMRI images were classified by the SVM supervised algorithm, to accurately map the spatial extent of all mangrove patches in the sites. Although mangrove mapping at the national scale, by analyzing manually selected multi-tidal images, was efficient in the present study, there is a challenge in collecting and selecting the appropriate tiles of Landsat multi-tidal data, especially in automatic approaches that might be devel-

oped on cloud computing platforms such as Google Earth Engine, so this challenge can be addressed in future research.

By employing the SMRI as a robust mangrove index on Landsat long-term data, the mangrove area of southern Iran was estimated at approximately 13,000 ha in 2020, showing an increase of 1471.1 ha from 1990 to 2000, a decrease of 427.4 ha from 2000 to 2010, and an increase of 1269.2 ha from 2010 to 2020. However, the low temporal resolution of the present study (i.e., three 10-year periods) prevented the investigation of the short-term (e.g., annual) trends of the habitat areas within the studied sites. The general increase in the mangrove area during the last three decades (i.e., approximately 2313 ha) can possibly be related to the establishment and management of national parks, protected areas, and biosphere reserves along the southern coast of the country, since the 1970s and after the Ramsar Convention on Wetlands (February 1971) held in Iran. Finally, we used the six most important landscape metrics for the description and quantification of the changes observed in Iran's mangrove forests during the study period. Our findings revealed stronger connectivity and higher complexity in most sites; however, the mangroves were fragmented and weakly connected within other sites. Anthropogenic activities such as severe conservation and afforestation have likely caused seaward and landward expansions of the forests within some sites, which may be addressed by future research.

The main achievement of this study was the accurate mapping of Iran's mangrove forests and their long-term dynamics, by means of a recently developed method, i.e., the SMRI from Landsat data. The mangrove forests of all study sites were mapped with a similar procedure for each study year, offering valuable insight into understanding the mangrove dynamics along the coast of southern Iran over time and space. Moreover, the robust mapping approach resulted in the identification of mangrove forests established in S8 after 2010, which had no mangroves before. However, the mangrove forest dynamics within the study sites, as reported in the present study, should be investigated regarding natural processes and anthropogenic activities in future.

**Author Contributions:** Conceptualization, Y.E. and M.L.N.; methodology, Y.E. and M.L.N.; software, M.L.N.; validation, Y.E. and M.L.N.; formal analysis, Y.E. and M.L.N.; investigation, Y.E., M.L.N. and K.S.; writing—original draft preparation, Y.E.; writing—review and editing, Y.E. and K.S. All authors have read and agreed to the published version of the manuscript.

**Funding:** This research was funded by the Iran National Science Foundation (INSF) under "The Project for Quantitative Assessment of Iran's Mangrove Forests by Remote Sensing Techniques" (Project number 98025568).

**Acknowledgments:** The authors are grateful to Uta Berger, full professor at Technical University of Dresden, Germany, for her constructive comments, suggestions, and encouragement.

**Conflicts of Interest:** The authors declare no conflict of interest. The funder had no role in the design of the study; in the collection, analyses, or interpretation of data; in the writing of the manuscript; or in the decision to publish the results.

## Appendix A

**Table A1.** Date of Landsat time series data used for each mangrove site in the present study.

| Mangrove Sites | Study Year | | Low-Tide Imagery Date | High-Tide Imagery Date |
|---|---|---|---|---|
| Bushehr Province | S1 | 1990 | 19 April 1990 | 25 August 1990 |
| | | 2000 | 13 March 2000 | 1 June 2000 |
| | | 2010 | 17 March 2010 | 20 May 2010 |
| | | 2020 | 30 October 2020 | 4 March 2020 |
| | S2 | 1990 | 27 March 1990 | 12 April 1990 |
| | | 2000 | 6 March 2000 | 7 April 2000 |
| | | 2010 | 11 April 2010 | 26 March 2010 |
| | | 2020 | 3 July 2020 | 13 March 2020 |

**Table A1.** *Cont.*

| Mangrove Sites | Study Year | | Low-Tide Imagery Date | High-Tide Imagery Date |
|---|---|---|---|---|
| Hormozgan Province | S3 | 1990 | 8 November 1990 | 13 March 1990 |
| | | 2000 | 11 May 2000 | 8 March 2000 |
| | | 2010 | 11 July 2010 | 12 March 2010 |
| | | 2020 | 28 February 2020 | 28 December 2020 |
| | S4 | 1990 | 8 November 1990 | 13 March 1990 |
| | | 2000 | 11 May 2000 | 8 March 2000 |
| | | 2010 | 11 July 2010 | 12 March 2010 |
| | | 2020 | 28 February 2020 | 28 December 2020 |
| | S5 | 1990 | 22 March 1990 | 3 December 1990 |
| | | 2000 | 7 March 2000 | 20 May 2000 |
| | | 2010 | 10 December 2010 | 29 March 2010 |
| | | 2020 | 12 June 2020 | 8 March 2020 |
| | S6 | 1990 | 6 March 1990 | 7 April 1990 |
| | | 2000 | 17 March 2000 | 27 October 2000 |
| | | 2010 | 10 December 2010 | 29 March 2010 |
| | | 2020 | 8 March 2020 | 18 October 2020 |
| | S7 | 1990 | 16 April 1990 | 15 March 1990 |
| | | 2000 | 10 March 2000 | 9 February 2001 |
| | | 2010 | 14 March 2011 | 30 March 2010 |
| | | 2020 | 1 March 2020 | 17 March 2020 |
| Sistan and Baluchestan Province | S8 | 1990 | 8 March 1990 | 27 May 1990 |
| | | 2000 | 3 March 2000 | 19 March 2000 |
| | | 2010 | 12 December 2011 | 31 March 2010 |
| | | 2020 | 10 March 2020 | 18 September 2020 |
| | S9 | 1990 | 28 November 1990 | 17 March 1990 |
| | | 2000 | 12 March 2000 | 25 December 2000 |
| | | 2010 | 15 December 2011 | 16 March 2010 |
| | | 2020 | 3 March 2020 | 6 May 2020 |

**Table A2.** The bands of Landsat 5, 7, and 8 data used to compute the vegetation and mangrove indices.

| Satellite and Sensor | Bands | Wavelength (Micrometer) | Spatial Resolution (m) |
|---|---|---|---|
| Landsat 5 TM | Band 1 | 0.45–0.52 | 30 |
| | Band 2 | 0.52–0.60 | 30 |
| | Band 3 | 0.63–0.69 | 30 |
| | Band 4 | 0.76–0.90 | 30 |
| | Band 5 | 1.55–1.75 | 30 |
| | Band 7 | 2.08–2.35 | 30 |
| Landsat 7 ETM+ | Band 1 | 0.45–0.52 | 30 |
| | Band 2 | 0.52–0.60 | 30 |
| | Band 3 | 0.63–0.69 | 30 |
| | Band 4 | 0.77–0.90 | 30 |
| | Band 5 | 1.55–1.75 | 30 |
| | Band 7 | 2.09–2.35 | 30 |
| Landsat 8 OLI | Band 2 | 0.45–0.51 | 30 |
| | Band 3 | 0.53–0.59 | 30 |
| | Band 4 | 0.64–0.67 | 30 |
| | Band 5 | 0.85–0.88 | 30 |
| | Band 6 | 1.57–1.65 | 30 |
| | Band 7 | 2.11–2.29 | 30 |

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
