# Peer review of "Assessment of Iran’s Mangrove Forest Dynamics (1990–2020) Using Landsat Time Series"

_remotesensing, doi:10.3390/rs14194912_

Round 1

Reviewer 1 Report

This paper aims to compare the efficiency of different vegetation indices in SVM classification, also monitors and analyzes the temporal and spatial changes on mangrove areas along the southern coast of Iran.

The following aspects about the classification algorithm requires to be improved:

A) Section 2.4 is suggested to be described in more details. It should explain what is the input parameters of SVM algorithm, only a single band of vegetation index or all the other image bands? Or does it include other combinations of mangrove features? Does Different combination of feature bands generate different classification accuracy?

B) Maybe it needs to validate the reasonability of accuracy evaluation in Section 2.5. In this study, AUC was selected as the accuracy measure for comparing the classification effect on different vegetation indices. It can be seen in Table 2 that the AUC of NDVI and SMRI are relatively high. However, most previous studies indicate that in the complex environment of coastal zone, NDVI as a single indicator cannot achieve a good classification result. Therefore, is it reasonable to take AUC as the measure to evaluate the classification accuracy of vegetation index?

C) Does it has clear meaning in the evaluation of SVM classification results with AUC? This comment is related to the previous one, that is, if the evaluation significance of AUC on SVM classification results is not clear or even meaningless, then higher AUC cannot verify that the classification result is more accurate. (AUC is calculated from ROC, and ROC curve is drawn from FPR and TPR under different threshold of probability, however, all the pixels distributed in the hyperplane of SVM will be outputted as binary classification type based on the model training, the location of hyperplane, which is determined by the training process, will vary with different training datasets, and probability threshold will of none sense in the drawing of the ROC curve? Detailed information can be referred to the website: https://qastack.cn/stats/37795/roc-curve-for-discrete-classifiers-like-svm-why-do-we-still-call-it-a-curve

In addition, the integrity of the article should be improved:

It is mentioned in the flow chart (line 389-395) of Section 2.5 and Figure 2 that the accuracy of the final mangrove map should be evaluated, but I cannot find this content in the later section.

Author Response

Reviewer 1

This paper aims to compare the efficiency of different vegetation indices in SVM

classification, also monitors and analyzes the temporal and spatial changes on mangrove areas along the southern coast of Iran.

The following aspects about the classification algorithm requires to be improved:

  1. A) Section 2.4 is suggested to be described in more details. It should explain what is the input

parameters of SVM algorithm, only a single band of vegetation index or all the other image

bands? Or does it include other combinations of mangrove features? Does Different combination

of feature bands generate different classification accuracy?

Section 2.4 was revised as suggested. As explained in the text, we considered single band of vegetation index as input to SVM to compare and explore the capability of each index in mangrove mapping. There is no doubt that if we add other bands (e.g., raw bands of Landsat images, other indices, bands of features such as homogeneity/entropy/etc) to SVM, it may improve the accuracy of mangrove mapping. However, we could not realize that the obtained accuracy would be related to the combination used or the evaluated index. Additionally, Baloloy et al. (2020) suggested threshold filtering to separate mangroves from non-mangroves on MVI images but we preferred to use a similar less subjective procedure to separate mangroves from other classes. We used SVM with constant parameters for all images of indices and we think the differences we observed in mangrove mapping accuracies were solely related to capability of indices.    

  1. B) Maybe it needs to validate the reasonability of accuracy evaluation in Section 2.5. In this study, AUC was selected as the accuracy measure for comparing the classification effect on different vegetation indices. It can be seen in Table 2 that the AUC of NDVI and SMRI are relatively high. However, most previous studies indicate that in the complex environment of coastal zone, NDVI as a single indicator cannot achieve a good classification result. Therefore, is it reasonable to take AUC as the measure to evaluate the classification accuracy of vegetation index?

Our results showed this fact that NDVI cannot provide an accurate map of LULC in areas with higher diversity of land covers as the lowest OA was observed in S2 with different land covers; however, the index was more accurate in S6 with less diversity of land covers. However, the NDVI was less effective compared to most MSIs in all sites.

We agree that this is theoretically true that ROC is not an appropriate index to evaluate classifiers like SVM. Therefore, we removed ROC and its AUC from the manuscript and replaced it by statistics of accuracy, precision, recall and f1-score.

  1. C) Does it has clear meaning in the evaluation of SVM classification results with AUC? This

comment is related to the previous one, that is, if the evaluation significance of AUC on SVM

classification results is not clear or even meaningless, then higher AUC cannot verify that the

classification result is more accurate. (AUC is calculated from ROC, and ROC curve is drawn

from FPR and TPR under different threshold of probability, however, all the pixels distributed in the hyperplane of SVM will be outputted as binary classification type based on the model training, the location of hyperplane, which is determined by the training process, will vary with different training datasets, and probability threshold will of none sense in the drawing of the ROC curve?

Detailed information can be referred to the website: https://qastack.cn/stats/37795/roc-curve-fordiscrete-classifiers-like-svm-why-do-we-still-call-it-a-curve

This comment was answered in the previous comment and we do agree with the reason that ROC is not suitable for this study. Moreover, we could not change the language of the link to English but we got the point.

In addition, the integrity of the article should be improved:

It is mentioned in the flow chart (line 389-395) of Section 2.5 and Figure 2 that the accuracy of the final mangrove map should be evaluated, but I cannot find this content in the later section.

We are sorry for this mistake that we forgot to provide the results of accuracy assessment. Table 3 was added to the text indicating the accuracy assessment of all SMRI-based mangrove maps in all sites.

Many thanks for the helpful points and comments

Reviewer 2 Report

This paper gives the brief spatio-temporal dynamics of Iran’s mangrove during 1990~2020. Some landscape metrics are validated. There are some comments as follow,

1) the contribution of this paper is not clear, all the indices or metrics are existing, the organizing of experiments is also regular;

2) the conclusion is not sufficient, whether the results of this paper novel? or the assessment is the most accurate?

3) the presented work in this paper is more like a standard processing flow, the author should declare the new points.

4) in the experiments, is there any cross validation with other data source? just like other optical satellite data, SAR data, ground truth, or Forest Department data.

5) there are a lot of high resolution satellite data (including high special resolution and spectral resolution), is it more suitable for mangrove monitoring?

6) for three decades assessment, there should be a general evolution trend for the mangrove in Iran, is there any principal factors leading the evolution?

The manuscript should be revised and improved. 

Author Response

Reviewer 2

This paper gives the brief spatio-temporal dynamics of Iran’s mangrove during 1990~2020. Some landscape metrics are validated. There are some comments as follow,

1) the contribution of this paper is not clear, all the indices or metrics are existing, the organizing of experiments is also regular;

We revised the manuscript (Last paragraph of Introduction and Conclusions) to show the main achievements of this study.

2) the conclusion is not sufficient, whether the results of this paper novel? or the assessment is the most accurate?

A paragraph was added to Conclusions to show the main novelty of this paper. The FRA2020 report indicated that Iran's mangrove forests were 25,760 ha in 1990, 2000, and 2010 with no changes! This was in contrast to other international publication like Makowski and Finkl [14] mentioned in the text. Therefore, we believe the findings may be important to international readers who are interested in global distribution of mangrove forests. Please see the response to comment 3.   

3) the presented work in this paper is more like a standard processing flow, the author should declare the new points.

Yes this is right; however, we think that this paper is novel from two points of view:

  1. Mangrove-specific indices have recently been developed for mangrove mapping (Since 2014) and their efficiency in sites with different environmental characteristics have less been investigated in contrast to well-known vegetation indices (such as NDVI). The performance of some of the indices (e.g., L8MI) has not been evaluated in other sites and we are not aware of their capabilities. Therefore, our study may help international researchers to better understand the performance of these indices in a different mangrove forest from where the indices have been developed. For example, Xia et al. (2020, Ecological Indicators) compared 6 indices (2 mangrove-specific and 4 vegetation indices) to map mangroves and it seems necessary to evaluate other indices such as L8MI and MVI.
  2. We do not have an accurate estimation of mangrove extent in southern Iran as one of the world mangrove forests grown on the northern border (latitude of 28 degrees) of mangrove extent. Even the status of Iran's mangrove forests reported in FAO documents (e.g., FRA2020) are not reliable since it was mentioned that Iran had 25760 ha of mangroves in 1990, 2000, and 2010 with no changes! Therefore, we think the present study can give a better insight to mangroves of Iran and their long-term dynamics that may be important to international readers.

The text was revised based on this comment.   

4) in the experiments, is there any cross validation with other data source? just like other optical satellite data, SAR data, ground truth, or Forest Department data.

Of course yes. We had field control points in some sites registered in March 2020 in S3, S4 and S7. In addition, we used aerial photographs of 1992 and 2006 with scale of 1:20,000 and 1:40,000, respectively; aerial images taken by UltraCam-D and Xp digital airborne cameras from 2014 to 2018 with approximate pixel size of 7 cm and Google Earth historical image data (from 1990 to 2020). Unfortunately, we do not have LiDAR data which is suitable for mangrove mapping.

5) there are a lot of high resolution satellite data (including high special resolution and spectral resolution), is it more suitable for mangrove monitoring?

Yes, this is right that there are other remote sensing data with higher spatial and spectral resolution such as WorldView. However, there are two main reasons that we used Landsat data in the present study:

  1. Landsat data is freely available to us in Iran that do not have access to other satellite data because of sanctions. We did our best to obtain data from Planet but we failed.
  2. One of our objectives was investigation of the long-term dynamics of mangrove forests in Iran and Landsat has the richest archive data (from 1973) compared to other satellite data, although it is not possible to use all of the images from 1973 in all parts of the world because of technical issues and atmospheric effects. We also have access to Sentinel data but the archive begins from 2015.      

6) for three decades assessment, there should be a general evolution trend for the mangrove in Iran, is there any principal factors leading the evolution?

To the best of our knowledge, mangrove forests in S3 and S4 were one of the first sites registered according to Ramsar Convention on Wetlands (February 1971) held in Iran. The documents from that time and other limited studies during these 50 years have shown that Iran's mangrove forests are purely covered by Avicennia marina. Only in small parts of two sites (S6 and S7), Rhizophora mucronata is mixed with A. marina. The results of previous published studies in Persian language and unofficial documents (such as newspaper reports) have shown no changes in species composition and biodiversity since 50 years ago in all sites. The only varying characteristic is their extent which is related to environmental and anthropogenic factors. Unfortunately, we suffer from a logical process of documenting all activities done by governmental and private organizations in mangrove forests. For example, we have spent a lot of time to collect unofficial reports on mangrove reforestation and afforestation from local offices to discuss the results of landscape metrics. Therefore, it is not easy to explain about the principal factors affecting mangroves in each separated site. We hope to find appropriate data to map other attributes of mangrove forests such as species composition in each site as a basis for investigation on evolution of these forests in future.   

The manuscript should be revised and improved. 

We did our best to consider the 6 comments when revising the manuscript.

Many thanks for the constructive comments

Reviewer 3 Report

The manuscript is a very well written paper on the important issue of Assessment of the  mangrove forest dynamics using Landsat time series. A very interesting study overall, with good science.

Author Response

Reviewer 3

The manuscript is a very well written paper on the important issue of Assessment of the mangrove forest dynamics using Landsat time series. A very interesting study overall, with good science.

Thank you very much for your positive comment.

Round 2

Reviewer 2 Report

The round 1 comments have been responded. I suggest that this manuscript can be published in RS.